# Wilson loops, holomorphic anomaly equations and blowup equations

Xin Wang[1*]

**1** Quantum Universe Center, Korea Institute for Advanced Study
* wxin@kias.re.kr

May 21, 2023

## Abstract

We investigate the topological string correspondence of the five-dimensional half-BPS Wilson loops on $S^1$. First, we propose the refined holomorphic anomaly equations for the BPS sectors of the Wilson loop expectation values. We then solve these equations and obtain many non-trivial novel integral refined BPS invariants for rank-one models. By studying the Wilson loop expectation values around the conifold point, we obtain the quantum spectra of the quantum Hamiltonians of the associated integrable systems. Lastly, as an application, the study of this paper leads to a generalization of the blowup equations for arbitrary magnetic fluxes that satisfying the flux quantization condition.

# 1   Introduction

Topological strings on non-compact Calabi-Yau threefolds (CY3's) are solvable and have significant connections to several important concepts in physics, such as supersymmetric gauge theories [1], Chern-Simons theories [2], matrix models [3], and integrable models [4].

The amplitudes of the topological strings on the non-compact Calabi-Yau threefold $X$ compute the BPS spectra of the corresponding compactified theory via compactifying M-theory on $X$. In this way, the low energy theory obtained from the geometric engineering [1,5] is a five-dimensional (5D) supersymmetric quantum field theory, which is a supersymmetric gauge theory or a non-Lagrangian theory with eight supercharges, on the background $\mathbb{R}^4 \times S^1$. In the low energy theory, the BPS particles are realized as the M2-branes wrapped on holomorphic two-cycles $C \in H_2(X, \mathbb{Z})$. They are characterized by non-trivial spins $(j_L, j_R)$ in the representation $SU(2)_L \times SU(2)_R = SO(4)$, which is the little group of massive particles in $\mathbb{R}^4 \times S^1$.

The correspondence between 5D gauge theory and topological string theory is extended to observables. One of the most important observables comes from the insertion of a three-dimensional half-BPS defect on $\mathbb{R}^2 \times S^1$. The defect partition function then is captured by the topological open strings, which can be obtained by inserting the Lagrangian submanifold on $X$ [6], counts the disk invariants [7]. They can be calculated from the refined Chern-Simons theories and the refined topological vertex [8–10].

Another interesting observable comes from the insertion of the half-BPS Wilson loop operator, which corresponds to the topological strings on the background $(X, \{C_i\})$, with an insertion of a sequence of *background non-compact primitive curves* $\{C_i\}$ [11]. The half-BPS Wilson loop operator in the 5D gauge theory on $\mathbb{R}^4 \times S^1$, which is winding around the time circle $S^1$, is defined by inserting the operator

$$W_{\mathbf{r}} = \text{Tr}_{\mathbf{r}} \mathcal{T} \exp\left( i \oint_{S^1} dt (A_0(t) - \varphi(t)) \right) \tag{1}$$

in the path integral formalism. Here $\mathcal{T}$ is the time ordering operator, $\mathbf{r}$ is a representation of the gauge group, $A_0(t) = A_0(\vec{x} = 0, t)$ is the zero component of the gauge field and $\varphi(t) = \varphi(\vec{x} = 0, t)$ is a scalar field that accompanies the gauge field to preserve half of the supersymmetries. The main object we want to study is the expectation values of such Wilson loop operators and their correspondence in topological string theories.

Consider a 5D $\mathcal{N} = 1$ supersymmetric gauge theory from M-theory compactification on a non-compact Calabi-Yau threefold $X$, the BPS partition function $Z_{\text{BPS}}$ can be defined and computed as a Witten index [12–17]. It counts the BPS states that are characterized by the spins $(j_L, j_R)$ from the rotation group $SO(4) \cong SU(2)_L \times SU(2)_R$ of massive particles. The free energy of the theory is then a generating function of the integral topological invariant $N_{j_L, j_R}^\beta$ which counts the number of particles with charge $\beta \in H_2(X, \mathbb{Z})$, mass $\beta \cdot t$ and spin $(j_L, j_R)$, written as a refined BPS expansion [9,18,19]

$$\mathcal{F}_{\text{BPS}} = \log Z_{\text{BPS}} = \sum_{\beta \in H_2(X, \mathbb{Z})} \sum_{j_L, j_R} (-1)^{2j_L + 2j_R} N_{j_L, j_R}^\beta \frac{\chi_{j_L}(k\epsilon_-) \chi_{j_R}(k\epsilon_+)}{k\left(q_1^{1/2} - q_1^{-1/2}\right)\left(q_2^{1/2} - q_2^{-1/2}\right)} e^{-k\beta \cdot t}, \tag{2}$$

where $\epsilon_{\pm} = \frac{1}{2}(\epsilon_1 \pm \epsilon_2)$, $q_{1,2} = e^{\epsilon_{1,2}}$ and $\chi_j$ is the $SU(2)$ character with highest weight $j$. The same BPS partition function can also be computed from the refined topological vertex method [9] and direct integration method [20–22] from the refined holomorphic anomaly equation, which provides explicit checks for the correspondence.

The insertion of the half-BPS Wilson loop operators has a D-brane realization [23]. The D-brane bound states provide field contents for the instanton calculations, that we can use localization and topological vertex method to compute the Wilson loop expectation values for the classical Lie groups [24–33] in various dimensions [1]. Consider a Wilson loop operator in the simplest representation which can be generated from a heavy, stationary source particle that carries electric charges. The source particle of the Wilson loop operator can also be understood as the M2-branes wrapped on a non-compact background primitive curve C. By doing so, we can deduce that the Wilson loop expectation value in the simplest representation has the BPS expansion [11, 34]

$$\left\langle W_{\mathbf{r}_C} \right\rangle = \sum_{\beta \in H_2(X,\mathbb{Z})} \sum_{j_L, j_R} (-1)^{2j_L + 2j_R} \widetilde{N}^{\beta}_{j_L, j_R} \chi_{j_L}(\epsilon_-) \chi_{j_R}(\epsilon_+) e^{-\beta \cdot t}, \tag{3}$$

in terms of non-negative integral *Wilson loop BPS invariants* $\widetilde{N}^{\beta}_{j_L, j_R}$. Higher representations of Wilson loops are obtained by adding a set of non-compact background primitive curves $\mathcal{S} = \{C_1, \cdots, C_n\}$ and the Wilson loop expectation values are written in terms of the BPS sectors $\mathcal{F}_{\mathrm{BPS}, \{C_i\}}$ [34],

$$\left\langle W_{\mathbf{r} = \mathbf{r}_1 \otimes \cdots \otimes \mathbf{r}_n} \right\rangle = \sum_{\{\mathcal{S}_{n_1}, \cdots, \mathcal{S}_{n_k}\} \in P_n(\mathcal{S})} \prod_{i=1}^{k} \mathcal{F}_{\mathrm{BPS}, \mathcal{S}_{n_i}}, \tag{4}$$

by summing over all the partitions $\{\mathcal{S}_{n_1}, \cdots, \mathcal{S}_{n_k}\} \in P_n(\mathcal{S})$ of the set $\mathcal{S}$. Each BPS sector has a BPS expansion and we refer the reader to Section 2 for more details.

In this paper, we will focus more on the study of the BPS sectors, the main result of this paper is that in the holomorphic limit, the genus $(n, g)$ BPS sectors satisfy the *refined holomorphic anomaly equations*

$$\frac{\partial}{\partial S^{ij}} \mathcal{F}^{(n,g)}_{\{C_1, \cdots, C_n\}} = \frac{1}{2} \left( D_i D_j \mathcal{F}^{(n,g-1)}_{\{C_1, \cdots, C_n\}} + \sum_{\substack{\mathcal{S}_{n'} \cup \mathcal{S}_{n-n'} = \mathcal{S}; \\ n'=0, \cdots, n}} \sum_{n', g'}{}' D_i \mathcal{F}^{(n', g'-n)}_{\mathcal{S}_{n'}} \cdot D_j \mathcal{F}^{(n-n', g-g'-n')}_{\mathcal{S}_{n-n'}} \right), \tag{5}$$

which are derived in Section 2. The refined holomorphic anomaly equations for the BPS sectors surprisingly coincide with those for the higher point correlation functions [35]. In Section 3, we specialize the discussion on local del Pezzo surfaces, which correspond to the 5D rank-one theories in the Coulomb branch.

The holomorphic anomaly equation can be used to solve the topological string amplitudes by using the direct integration method [20–22], and up to holomorphic ambiguities, the amplitudes can be solved entirely. The holomorphic ambiguities can be fixed by using the gap conditions [36, 37], and other possible boundary conditions. In this paper, we use the direct integration method to solve the BPS sectors for local $\mathbb{P}^1 \times \mathbb{P}^1$ and local $\mathbb{P}^2$, we proposed the boundary conditions of the BPS sectors, which makes it possible to compute the BPS invariants of Wilson loops to arbitrary genera in arbitrarily high representations. In Section 4, we discuss the magnetic dual of the Wilson loop, which corresponds to the expansion of the B-model Wilson loop expectation value around the conifold point. In particular, we recover the quantized spectrum of the corresponding quantum Hamiltonians of the integrable systems.

---

[1] In 4D, they are chiral operators; in 6D, they are Wilson surface operators.

In Section 5, we revisit the B-model of the blowup equations. The blowup equation is another powerful tool for solving the BPS partition functions. It was first derived in the 4D and 5D supersymmetric gauge theories [38–40] and was later generalized to topological string theories [41]. The blowup equation of a given theory is classified by the magnetic fluxes, which were usually thought to be bounded. As we will see in Section 5, for arbitrary magnetic fluxes that satisfy the flux quantization condition, the blowup equations are still valid but there will be generically dependencies of the expectation values of the Wilson loops. Our findings give a large class of generalization to the blowup equations.

The paper is organized as follows. In Section 2, we review the Wilson loops correspondence in topological string theory and derive the refined holomorphic anomaly equations for the BPS sectors. In Section 3, we discuss BPS sectors for local del Pezzo surfaces. By using the direct integration method in the B-model, we explicitly compute the BPS sectors to very high representations. In Section 4, based on the B-model expression of the BPS sectors, we study the Wilson loop expectation values around the conifold point from which can be used to derive the quantum spectra of the corresponding integrable systems. In Section 5, we discuss the application of the Wilson loop in the blowup equations. In particular, we generalize the blowup equation for arbitrary magnetic fluxes that satisfy the flux quantization conditions. Section 6 provides the conclusions of this paper. In Appendix A, we review the E-string partition function from the refined topological vertex method and in Appendix B we derive the Wilson loop expectation values for $D_5, E_6, E_7, E_8$ del Pezzo's in the fundamental representation from the E-string partition function. In Section C, we provide the Wilson loop BPS invariants.

## 2 Wilson loops and topological strings

This section gives a general description of the Wilson loops and topological strings correspondence. Most of the content here can be found in [34], but we will also introduce new things. We will first introduce the notations of the refined topological strings and the Wilson loops. We then describe the BPS expansion of the Wilson loop expectation values involving the BPS sectors. Then based on the refined holomorphic anomaly equations for the Wilson loop amplitudes first introduced in [34], we derive the refined holomorphic anomaly equation for the BPS sectors. Our main results of this section are included in equation (22) and (23).

**The refined topological strings**   The refined topological strings have been extensively studied and there are many pedagogical introductions on this topic, e.g. [42]. In this subsection, we give a brief description of the notations for later use.

The refined A-model topological strings on the Calabi-Yau threefold $X$ compute the refined BPS invariants of the corresponding five-dimensional gauge theory with eight supercharges, the whole partition function can be formally written as

$$Z_{\text{top}}(\epsilon_1, \epsilon_2, t) \equiv e^{\mathcal{E}} Z_{\text{BPS}}, \tag{6}$$

or simply denoted as $Z(\epsilon_1, \epsilon_2, t)$, where $t$ is the Kähler parameter, $Z_{\text{BPS}}$ is the BPS part defined in (2), and $\mathcal{E}$ is the singular part that can be written in terms of the classical geometrical invariants

$$\mathcal{E} = \frac{1}{6\epsilon_1 \epsilon_2} a_{ijk} t_i t_j t_k + b_i^{(0,1)} t_i + \frac{(\epsilon_1 + \epsilon_2)^2}{\epsilon_1 \epsilon_2} b_i^{(1,0)} t_i, \tag{7}$$

where $a_{ijk}$ are the triple intersection numbers and $b_i^{(1,0)}$ and $b_i^{(1,0)}$ can also be determined from the geometric information. For a recent discussion, see [43]. We will also define the free

energies or the amplitudes

$$\mathcal{F}(\epsilon_1, \epsilon_2, t) = \log Z_{\text{top}}(\epsilon_1, \epsilon_2, t) = \sum_{n,g=0}^{\infty} (\epsilon_1 + \epsilon_2)^{2n} (\epsilon_1 \epsilon_2)^{g-1} \mathcal{F}^{(n,g)}(t). \tag{8}$$

**The Wilson loops**    The expectation values of the half-BPS Wilson loop operators in the Coulomb branch of a 5D $\mathcal{N} = 1$ gauge theory on $\mathbb{R}^4_{\epsilon_1, \epsilon_2} \times S^1$ are generated by heavy stationary quarks, which can be obtained by inserting the background non-compact curves $\{C_1, \cdots, C_n\}$ in the Calabi-Yau geometry $X$, that was first introduced in [11] and further studied in [34]. We refer to these curves as *primitive curves* if they individually generate Wilson loops in non-decomposable representations $\mathbf{r}_{C_i}$ or in short $\mathbf{r}_i$ of the gauge group. The multiple primitive curves $\{C_1, \cdots, C_n\}$ insertion generates the Wilson loop in the tensor product of representation $\mathbf{r} = \mathbf{r}_1 \otimes \cdots \otimes \mathbf{r}_n$.

The BPS partition function $Z_{W_{\mathbf{r}}}(\epsilon_1, \epsilon_2, t)$ with the insertion of Wilson loop operator $W_{\mathbf{r}}$ can be obtained by computing the topological string amplitudes on the background $(X, \{C_i\})$. We denote $Z(\epsilon_1, \epsilon_2, t)$ without the subscript $_{W_{\mathbf{r}}}$ as the BPS partition of the 5D gauge theory, or equivalently the topological string partition function on $X$, then the expectation value of the Wilson loop operator can be expressed as

$$\langle W_{\mathbf{r}} \rangle = \frac{Z_{W_{\mathbf{r}}}(\epsilon_1, \epsilon_2, t)}{Z(\epsilon_1, \epsilon_2, t)}, \tag{9}$$

which is the main object we study in this paper.

**The BPS expansion**    From the M-theory perspective, the M2-branes wrapped on the non-compact primitive curves $\{C_1, \cdots, C_n\}$ provide stationary heavy quarks in the 5D gauge theory. If we first treat these quarks as dynamic particles with masses $m_i$, their BPS spectrum can be written as the refined Gopakumar-Vafa expansion originated from [18, 19] and extensively studied in [44]. Then we consider them as non-dynamic background particles, their masses are defined by the effective masses $\widetilde{M}_i \equiv \mathcal{I}^{-1} \cdot e^{-m_i}$ by absorbing the momentum factor

$$\mathcal{I} = 2\sinh(\epsilon_1/2) \cdot 2\sinh(\epsilon_2/2), \tag{10}$$

to remove the dynamic degrees of freedom. As discussed in [34], the Wilson loop expectation value can be computed from the generating function

$$Z_{\text{gen}} = \exp\left( \sum_{\{C_{l_{i,1}}, \cdots, C_{l_{i,n_i}}\} \in \mathcal{P}(\mathcal{S})} \mathcal{F}_{\text{BPS}, \{C_{l_{i,1}}, \cdots, C_{l_{i,n_i}}\}} \cdot M_{l_{i,1}} \cdots M_{l_{i,n_i}} \right), \tag{11}$$

by summing over the power set $\mathcal{P}(\mathcal{S})$ of the set of primitive curves $\mathcal{S} = \{C_1, \cdots, C_n\}$, where the power set $\mathcal{P}(\mathcal{S})$ is all the subsets of $\mathcal{S}$, including the empty set and $\mathcal{S}$ itself. Then the expectation value of the Wilson loop operator in the representation $\mathbf{r} = \mathbf{r}_1 \otimes \cdots \otimes \mathbf{r}_n$ can be derived by considering the coefficient of $\prod_{i=1}^n \widetilde{M}_i$ in the generating function $Z_{\text{gen}}$ in the heavy mass limit $m_i \to \infty$. We obtain the BPS expansion

$$\left\langle W_{\mathbf{r} = \mathbf{r}_1 \otimes \cdots \otimes \mathbf{r}_n} \right\rangle = \sum_{\{\mathcal{S}_{n_1}, \cdots, \mathcal{S}_{n_k}\} \in P_n(\mathcal{S})} \prod_{i=1}^{k} \mathcal{F}_{\text{BPS}, \mathcal{S}_{n_i}}, \tag{12}$$

where we define $P_n(\mathcal{S})$ as the partition of the set $\mathcal{S}$ with elements $\{\mathcal{S}_{n_1}, \cdots, \mathcal{S}_{n_k}\} \in P_n(\mathcal{S})$, each element $\mathcal{S}_{n_i} = \{C_{l_{i,1}}, \cdots, C_{l_{i,n_i}}\}$ is a set of $n_i$ primitive curves. $\mathcal{F}_{\text{BPS}, \{C_1, \cdots, C_n\}}$ is defined as the

BPS sector in the representation $\mathbf{r} = \mathbf{r_1} \otimes \cdots \otimes \mathbf{r_n}$. It has the BPS expansion [2]

$$\mathcal{F}_{\text{BPS},\{C_1,\cdots,C_n\}} = \mathcal{I}^{n-1} \cdot \sum_{\beta \in H_2(X,\mathbb{Z})} \sum_{j_L,j_R} (-1)^{2j_L+2j_R} \widetilde{N}^{\beta}_{j_L,j_R} \chi_{j_L}(\epsilon_-) \chi_{j_R}(\epsilon_+) e^{-\beta \cdot t}, \qquad (13)$$

in terms of the *refined Wilson loop BPS invariants* $\widetilde{N}^{\beta}_{j_L,j_R}$, which counts the number of BPS particles of spin $(j_L, j_R)$. In particular, when the number of primitive curves n equals zero, the BPS sector captures the conventional refined topological string amplitudes (8).

**The refined holomorphic anomaly equations**   The free energies $\mathcal{F}^{(n,g)}$ defined in (8) satisfy the refined holomorphic anomaly equations which have been proposed in [22,45], as a refined generalization of the work of BCOV [46]. In the holomorphic limit, they read

$$\frac{\partial \mathcal{F}^{(n,g)}}{\partial S^{ij}} = \frac{1}{2}\left( D_i D_j \mathcal{F}^{(n,g-1)} + {\sum_{n',g'}}' D_i \mathcal{F}^{(n',g')} \cdot D_j \mathcal{F}^{(n-n',g-g')} \right), \quad n+g > 1, \qquad (14)$$

where the prime in the summation means the omission of $(n', g') = (0, 0)$ and $(n, g)$. Here $S^{ij}$ is the propagator, and $D_i$ is the covariant derivative. The (refined) holomorphic anomaly equations provide a very powerful method, which is usually called the direct integration method, to compute the topological string amplitudes in the B-model. The direct integration method states that by integrating over the propagators $S^{ij}$ on both sides of the holomorphic anomaly equation (14), up to holomorphic ambiguities, we can solve the topological string amplitudes genus by genus recursively. For more notations and descriptions, please refer to the textbook [42], as we do not include all the details here.

In [34], it was further observed that for any representation $\mathbf{r}$, by defining the *Wilson loop amplitudes*

$$\mathcal{W}_{\mathbf{r}} = \log\langle W_{\mathbf{r}} \rangle = \sum_{n,g=0}^{\infty} (\epsilon_1 + \epsilon_2)^{2n} (\epsilon_1 \epsilon_2)^{g-1} \mathcal{W}_{\mathbf{r}}^{(n,g)}, \qquad (15)$$

the combinations

$$\mathcal{G}_{\mathbf{r}}^{(n,g)} \equiv \mathcal{F}^{(n,g)} + \mathcal{W}_{\mathbf{r}}^{(n,g)}, \qquad (16)$$

which are the free energies of the whole Wilson loop partition function $Z_{W_{\mathbf{r}}}$, satisfy the same refined holomorphic anomaly equations

$$\frac{\partial \mathcal{G}_{\mathbf{r}}^{(n,g)}}{\partial S^{ij}} = \frac{1}{2}\left( D_i D_j \mathcal{G}_{\mathbf{r}}^{(n,g-1)} + {\sum_{n',g'}}' D_i \mathcal{G}_{\mathbf{r}}^{(n',g')} \cdot D_j \mathcal{G}_{\mathbf{r}}^{(n-n',g-g')} \right), \quad n+g > 1. \qquad (17)$$

Based on (17), many calculations have been done for various models in [34], including models like local $\mathbb{P}^2$ which does not have a gauge theory correspondence. However, at least at this moment, the physical understanding of the amplitudes $\mathcal{G}_{\mathbf{r}}^{(n,g)}$ are still unclear, even though they also have the refined BPS expansions in terms of integral refined BPS invariants (but could be negative).

---

[2]It is also possible to expand the BPS sector in the refined Gopakumar-Vafa expansion

$$\mathcal{F}_{\text{BPS},\{C_1,\cdots,C_n\}} = \mathcal{I}^{n-1} \cdot \sum_{\beta \in H_2(X,\mathbb{Z})} \sum_{g_L,g_R} (-1)^{2g_L+2g_R} \widetilde{\text{GV}}^{\beta}_{g_L,g_R} (2\sinh(\epsilon_-/2))^{2g_L} \cdot (2\sinh(\epsilon_+/2))^{2g_R} e^{-\beta \cdot t},$$

where $\epsilon_{\pm} = \frac{1}{2}(\epsilon_1 \pm \epsilon_2)$.

To solve the refined holomorphic anomaly equations, we use the direct integration method, with the holomorphic ambiguities in the following ansatz

$$f_{n,g}(z) = \sum_{i=1}^{\delta} \sum_{k=0}^{o(i)} \frac{p_i^{(k)}}{\Delta_i^k}, \tag{18}$$

where $\delta$ is the number of components $\Delta_i$ of the discriminant, $o(i)$ gives the maximal singularity that one has at the corresponding type of divisor and $p_i^{(k)}$ is a polynomial of $z_i$. In particular for the conifold divisors $o(i) = 2(n+g) - 2$. If at the orbifold point where $\frac{1}{z_i} \to 0$, the amplitudes are regular, then the degrees of $p_i^{(k)}$ are generically bounded with the highest degree

$$o(i) \times \mathrm{ord}(\Delta_i) + \sigma\, o(i), \tag{19}$$

with a shift $\sigma$.

For the cases of local $\mathbb{P}^2$ and local $\mathbb{P}^1 \times \mathbb{P}^1$ without insertion of Wilson loops, it was observed [21,47] that the refined topological string amplitudes $\mathcal{F}^{(n,g)}$ are regular at the orbifold point, together with the *gap condition* that was derived in [37] from the Schwinger integral representation of the Gopakumar-Vafa expansion, we can fix the holomorphic ambiguities completely thus solve the refined topological string amplitudes to any high enough genus $(n,g)$. For the Wilson loop amplitudes $\mathcal{W}_{\mathbf{r}}^{(n,g)}$, even though they are completely regular at the conifold point, it has been noticed in [34] that they are not regular when the representations are large, so that we can not fix the holomorphic anomaly completely even for local $\mathbb{P}^2$ and local $\mathbb{P}^1 \times \mathbb{P}^1$ without additional inputs of boundary conditions. In the next subsection, we will see that if we consider the holomorphic anomaly equation for the BPS sectors, all of the disadvantages are resolved.

**The refined holomorphic anomaly equations for the BPS sectors**    We first define the partition function

$$Z_{\mathrm{HAE}} = \exp\left(\mathcal{F}_{\mathrm{HAE}}\right) = \exp\left( \sum_{n+g \geq 1} (\epsilon_1 + \epsilon_2)^{2n} (\epsilon_1 \epsilon_2)^{g-1} \mathcal{F}^{(n,g)} \right), \tag{20}$$

which is the topological string partition function, by setting the genus zero contribution to zero. It is not difficult to demonstrate that the refined holomorphic anomaly equations (14) can be rewritten in the form of the heat kernel equation [46]

$$\left[ \frac{\partial}{\partial S^{ij}} - \frac{\epsilon_1 \epsilon_2}{2} D_i D_j \right] Z_{\mathrm{HAE}} = 0, \tag{21}$$

in the holomorphic limit. Similarly, one can derive that the combination $\langle W_{\mathbf{r}} \rangle Z_{\mathrm{HAE}}$ satisfies the same equation, by combining it with (21), we can derive the holomorphic anomaly equation

$$\frac{\partial}{\partial S^{ij}} \langle W_{\mathbf{r}} \rangle = \frac{\epsilon_1 \epsilon_2}{2} \left( D_i D_j \langle W_{\mathbf{r}} \rangle + D_i \mathcal{F}_{\mathrm{HAE}} D_j \langle W_{\mathbf{r}} \rangle + D_j \mathcal{F}_{\mathrm{HAE}} D_i \langle W_{\mathbf{r}} \rangle \right). \tag{22}$$

Substituting the BPS expansion (12) in the holomorphic anomaly equation (22), we can derive the refined holomorphic anomaly equations for the BPS sectors for the primitive curves class $\mathcal{S} = \{C_1, \cdots, C_n\}$

$$\frac{\partial}{\partial S^{ij}} \mathcal{F}_{\{C_1, \cdots, C_n\}}^{(n,g)} = \frac{1}{2} \left( D_i D_j \mathcal{F}_{\{C_1, \cdots, C_n\}}^{(n,g-1)} + \sum_{\substack{\mathcal{S}_{n'} \cup \mathcal{S}_{n-n'} = \mathcal{S} \\ n'=0,\cdots,n}} {\sum_{n',g'}}' D_i \mathcal{F}_{\mathcal{S}_{n'}}^{(n',g'-n)} \cdot D_j \mathcal{F}_{\mathcal{S}_{n-n'}}^{(n-n',g-g'-n')} \right). \tag{23}$$

Here the first sum on the right-hand side sum over all the subsets $\mathcal{S}_{n'}$ and $\mathcal{S}_{n-n'}$ of the primitive curve class $\mathcal{S}$, with the length of the subsets to be $n'$ and $n-n'$ respectively. The prime on the second sum means we sum over all the integers $0 \leq n' \leq n$, $0 \leq g' \leq g+n$ by excluding $n'+g'=0$ and $n'+g'=n+g+n$. The set $\mathcal{S}_{n'}$ or $\mathcal{S}_{n-n'}$ can be empty, when it is empty, we define the corresponding amplitudes to be the conventional refined topological string amplitudes (8)

$$\mathcal{F}^{(n,g)}_{\mathcal{S}=\{\}} \equiv \mathcal{F}^{(n,g)}. \tag{24}$$

For any negative genus, we use the notation

$$\mathcal{F}^{(n,g<0)}_{\{\mathsf{C}_1,\cdots,\mathsf{C}_n\}} = 0. \tag{25}$$

The refined holomorphic anomaly equations for the BPS sectors are valid for any genus $(n,g)$ with $n+g >= 0$. When the set of the primitive curves is empty, these equations are reduced to the conventional holomorphic anomaly equations of refined topological string that are valid for $n+g > 1$.

From the refined holomorphic anomaly equation for the BPS sectors (23), we can use the direct integration method to compute the BPS sectors from the lower genus to the higher genus and from the low number of primitive curves to the higher number of primitive curves recursively. But we need to slightly change the form of the holomorphic ambiguities due to the asymptotic behavior at the large volume point $t \to \infty$. For models such as local $\mathbb{P}^2$ and local $\mathbb{P}^1 \times \mathbb{P}^1$, we can verify that the BPS sectors are regular at both conifold point and orbifold point. The regularity condition provides enough boundary conditions to fix the holomorphic ambiguities that make it possible to solve the BPS sectors for arbitrary genera and arbitrary numbers of primitive curves for these two models. We will show the detailed calculations in Section 3.3.

**Additive property** The refined holomorphic anomaly equations for the Wilson loop expectation values (22) are completely linear. This means any linear combination of Wilson loop expectation values of different representations, with coefficients that can be arbitrary functions of $\epsilon_1, \epsilon_2$ or even mass parameters $m_i$, will still satisfy the same refined holomorphic anomaly equation in the form (22).

**Initial conditions** Suppose we know all the refined topological string amplitudes $\mathcal{F}^{(n,g)}$, the direct integration method still requires some initial conditions of the recursion equation (23), that is we need the explicit expression of the genus zero amplitudes of the BPS sectors of a single primitive curve. The key information has been discussed in [34], here we give a more generic discussion to arbitrary non-compact Calabi-Yau threefolds [3].

Consider a non-compact Calabi-Yau threefold $X$, which is not necessarily toric. We keep in mind that the low energy physics are described by a 5D gauge theory but the discussion here is valid for theory without a gauge theory description. We denote by $b_2$ and $b_4$ the Betti numbers that count the number of independent compact divisors and independent curves of $X$ respectively. In general, for a non-compact Calabi-Yau manifold, $b_2 \geq b_4$, so the Kähler parameters $t_i$, which are the volume of the curves in $X$, can be divided into Coulomb parameters $\alpha_i$ and mass parameters $m_i$ in the language of gauge theory. We have

$$t = (\alpha_1, \cdots, \alpha_{b_4}, m_1, \cdots, m_{b_2-b_4}), \tag{26}$$

and $\alpha_i$ are also called "true" Kähler parameters in some literature [41], they are supposed to be the coordinates that all the degrees of $\alpha_i$ in the BPS expansion (2) are non-negative. We will

---

[3]More precisely, these are arbitrary non-compact Calabi-Yau threefolds for which there is no non-shrinkable curve.

refer to the curves that correspond to the Coulomb parameters as compact curves and denote $z_i$ to be the related complex structures in the B-model, we usually have the mirror maps in the expansion

$$\alpha_i = \log(z_i) + \mathcal{O}(z_i), \tag{27}$$

at the large volume limit $z_i \to 0$. We define $-C_{ij}$ as the intersection matrix between the compact divisors and compact curves; when there is a gauge theory description, $C_{ij}$ is the Cartan matrix of the 5D gauge group.

Finally, for the Calabi-Yau manifold $X$ we have described above, there are $b_4$ independent non-decomposable representations whose highest weights are the fundamental weights. Instead of the representations, we consider the orbits generated from these fundamental weights and denoted the Wilson loop operators as $W_{\mathbf{r}_i}$, which are generated from the primitive curves $C_i$, for $i = 1, \cdots, b_4$. All other primitive curves are isomorphic to these curves. We call $b_4$ to be the rank of theory, as the number of independent Coulomb parameters. For each Wilson loop operator $W_{\mathbf{r}_i}$, the expectation value is equal to the amplitudes of the BPS sector $\mathcal{F}_{C_i}$, at genus zero they have the value

$$\mathcal{F}_{C_i}^{(0,0)} = \prod_{j=1}^{b_4} z_j^{-C_{ij}^{-1}}, \qquad i = 1, \cdots, b_4. \tag{28}$$

In the large volume limit $\prod_{j=1}^{b_4} z_j^{C_{ij}^{-1}} \to 0$, the genus zero part of the BPS sector (28) is singular, but we hope that the higher genus parts are regular under the large volume limit as the initial conditions for the higher genus BPS sectors. Such a requirement is equivalent to fixing the coefficient of leading order expansion of the BPS sector to be one. Then the higher genus amplitudes and the amplitudes with the insertion of more primitive curves can be obtained from the initial conditions (28) together with the refined holomorphic anomaly equation and some other inputs of boundary conditions.

Note that the initial conditions here only solve the orbit of the fundamental weight of the gauge group. The Wilson loop of a representation can be obtained by linear combinations of these orbits, according to the additive property of the refined holomorphic anomaly equation.

Lastly, we emphasize that the initial conditions for the BPS sectors are completely the choice by hand. One can choose other initial conditions but the final result of the Wilson loop expectation values should be the same, up to an irrelevant factor, due to the additive property of the refined holomorphic anomaly equation (22).

## 3 Wilson loops for del Pezzo surfaces

In this section, we specialize the descriptions and calculations of Wilson loop expectation values for the cases of local del Pezzo surfaces. We will take the massless limit, resulting in only one modulus in both the topological string A-model and B-model. These are denoted as $t$ and $z$ respectively.

### 3.1 Topological strings on local del Pezzo surfaces

The topological strings on a local del Pezzo surface $X$, which is given by the anti-canonical bundle over the del Pezzo surface $S$, provides the BPS spectrum of a 5D $\mathcal{N} = 1$ rank-one supersymmetric quantum field theory that is obtained from the M-theory compactification on $X$ [48]. Del Pezzo surfaces, which are finitely classified, are smooth, projective algebraic surfaces with ample anticanonical bundles. They can be described by $\mathbb{P}^1 \times \mathbb{P}^1$ and $n$-point

blowups of $\mathbb{P}^2$ up to $n = 8$, which gives the local Calabi-Yau threefolds which are called local $\mathbb{P}^1 \times \mathbb{P}^1$ and $dP_n$. When $n > 1$, the $n$-point blowups of $\mathbb{P}^2$ are isomorphic to the $(n-1)$-point blowups of $\mathbb{P}^1 \times \mathbb{P}^1$, where the latter describes the 5D low energy theory with gauge group $SU(2)$ and $N_f = (n-1)$ hypermultiplets transforming in the fundamental representation of $SU(2)$. These rank-one theories have enhanced global symmetry $E_n$, so when $n = 5, 6, 7, 8$, we also name them as $E_6 = D_5, E_7, E_8$ theories and we refer to the corresponding geometries as $D_5, E_7, E_8$ del Pezzo's.

For the 9-point blowups of $\mathbb{P}^2$, the corresponding Calabi-Yau threefold is called $dP_9$ or the local half K3 surface, the corresponding lower-dimensional theory is no longer a 5D theory but rather a 6D theory which is known as E-string theory. For E-strings, the global symmetry becomes the affine Lie group $E_8^{(1)}$, so we will also use $E_8^{(1)}$ to denote the theory.

**Wilson loops from the heavy mass limit**    Let's first consider the case $dP_n$ with $n \leq 9$. We denote $h$ as the curve that is associated with the original $\mathbb{P}^2$ and $e_i$ the exceptional curve that is associated with the $i$-th blowup, they have the non-vanishing intersection numbers $h^2 = -e_i^2 = 1$. Then the Kähler parameters can be described by $t$ which is proportional to the volume of $h$ and $m_i$ which is the volume of $e_i$ with a shift

$$t = \frac{1}{3}\text{Vol}(h), \qquad m_i = \text{Vol}(e_i) - \frac{1}{3}\text{Vol}(h), \quad i = 1, \cdots, n. \tag{29}$$

Denote $Z^{dP_n}(t, m_1, \cdots, m_n, \epsilon_1, \epsilon_2)$ as the partition function of the refined topological strings on $dP_n$, in the Calabi-Yau phase that all the degree of the curves $e_i$ have positive degrees. For an integer $\mathsf{n} \leq 9 - n$, the heavy mass limit of the partition function on $dP_{n+\mathsf{n}}$ can be regarded as a generating function (11) of the Wilson loops according to the expansion [4]

$$Z^{dP_{n+\mathsf{n}}}(t, m_1, \cdots, m_{n+\mathsf{n}}, \epsilon_1, \epsilon_2) = Z^{dP_n}(t, m_1, \cdots, m_n, \epsilon_1, \epsilon_2)\left(1 + \left\langle W_{[-1]^{\otimes \mathsf{n}}}^{dP_n} \right\rangle \prod_{i=1}^{\mathsf{n}} \widetilde{M}_i + \cdots \right), \tag{31}$$

where we use $\cdots$ to denote all other contributions and $\widetilde{M}_i$ are the effective masses

$$\widetilde{M}_i = \frac{e^{-m_{n+i}}}{2\sinh(\epsilon_1/2) \cdot 2\sinh(\epsilon_2/2)}, \quad i = 1, \cdots, \mathsf{n}. \tag{32}$$

The coefficient $\left\langle W_{[-1]^{\otimes \mathsf{n}}}^{dP_n} \right\rangle$ then is supposed to be the Wilson loop expectation value of the model $dP_n$ in the representation $[-1]^{\otimes \mathsf{n}}$ with $n + \mathsf{n} \leq 9$ where $[-1]^{\otimes \mathsf{n}}$ $\mathsf{n}$-th tensor product of the "representation" $[-1]$ which means

$$\left\langle W_{[-1]^{\otimes \mathsf{n}}}^{dP_n} \right\rangle = e^{-\frac{\mathsf{n}}{3}t}(1 + \mathcal{O}(e^t)). \tag{33}$$

In the self-dual case with $\epsilon_1 = -\epsilon_2 = g_s$, the partition function of topological strings is reduced to the partition function of conventional topological strings, which capture the Gromov-Witten invariants of the Calabi-Yau threefold $dP_n$. Then for $n + \mathsf{n} \leq 9$, equation (31) can be treated as

---

[4]The Wilson loop expectation value can also be obtained from a similar expansion of the partition function with codimension-two defects that live on $\mathbb{R}_{\epsilon_1}^2 \times S^1$ and fixed on the other directions $\mathbb{R}_{\epsilon_2}^2$. The codimension-two defects are realized as refined topological branes [10, 49] in the topological string theory. Let's denote the $X$ to be the defect parameter, the BPS particles that are related to the defects that are only movable along half of the Omega-deformed space, so the effective mass to expand the partition function should be

$$\widetilde{X} = \frac{X}{2\sinh(\epsilon_1/2)}. \tag{30}$$

a mathematically rigorous way of defining the Gromov-Witten invariants of the Wilson loop. For the same reason, in the refined case, if $n + \mathsf{n} \leq 9$, the refined Wilson loop BPS invariants can be connected to the refined stable pair invariants [50]. However, there is no bound for the representation of a Wilson loop operator, it is interesting to find a direct mathematical definition of the Gromov-Witten invariants of the Wilson loops for arbitrary representations $[-1]^{\otimes \mathsf{n}}$.

To have a better understanding of the expansion (31), we provide an example. In [47], the massive refined BPS invariants for E-strings are computed. The geometry of the E-string theory can be treated as an elliptic-fibered CY3. Denoting $t$ and $\tau$ as the base and fiber parameters of the elliptic fibration, and other mass parameters are represented as the characters of $E_8$ group. For the curve class that the degrees of $t$ and $\tau$ are $d = (d_1, d_2)$, the first few refined BPS invariants $\oplus [N^d_{j_L, j_R}; (j_L, j_R)]$ are computed in [47]. If $d = (1, 0)$, we have

$$[\mathbf{1}; (0, 0)], \tag{34}$$

if $d = (1, 1)$, they are

$$[\mathbf{248}; (0, 0)] \oplus [\mathbf{1}; (\frac{1}{2}, \frac{1}{2})], \tag{35}$$

if $d = (1, 2)$, they are

$$[\mathbf{3875} + \mathbf{248} + 2 \times \mathbf{1}; (0, 0)] \oplus [\mathbf{248} + \mathbf{1}; (\frac{1}{2}, \frac{1}{2})] \oplus [\mathbf{1}; (1, 1)]. \tag{36}$$

In the limit to $E_8$ del Pezzo, we take the limit $\tau \to \infty$, but keep $t^{E_8} = t + \tau$ finite. To make the limit finite, we need to flip the degree $d = (1, 0)$ to $d = (-1, 0)$ in the E-string geometry, after doing this, the leading order coefficients of the $\tau$ parameter should be the refined Wilson loop BPS invariants for $E_8$ del Pezzo surface. So the invariants (34) and (36) are the degree $-1$ and degree $1$ invariants for the Wilson loop of $E_8$ model. In this way, we compute the refined Wilson loop BPS invariants for $D_5, E_6, E_7, E_8$ models, from the known refined BPS invariants of E-strings. We list them in Appendix C.3 in the massless case.

Note that the Wilson loops can be computed from the topological strings on the background $(X, \{\mathsf{C}_i\})$ with a set of primitive curves. The $\mathsf{n}$-point blowups of $dP_n$ provides an embedding geometry for the background $(X, \{\mathsf{C}_i\})$. However, the embedding geometry is not unique. For example, one can consider the genus-one fibered CY3 over a $-1$ curve, with fiber types $D_5, E_6, E_7$ [51, 52]. By selecting the next-to-leading order coefficients of $e^{-\tau}$, we can also obtain the refined Wilson loop BPS invariants for $D_5, E_6, E_7$ del Pezzo's in the fundamental representations.

As we have addressed, when $n > 1$, the $(n + 1)$-point blow up of $\mathbb{P}^2$ is isomorphic to the $n$-point blow up of $\mathbb{P}^1 \times \mathbb{P}^1$. But the latter description is more suitable for the corresponding 5D $SU(2)$ gauge theory with $N_f = n$ fundamental flavors. Recently, a refined topological vertex formalism for 5D $SU(2)$ theory with eight fundamental hypermultiplets was proposed in [53], which is reviewed in Appendix A. The theory is the Kaluza-Klein (KK) theory of E-string theory compactified on a circle. Following the logic above, we explicitly derive the partition functions of the Wilson loop operators in the fundamental representations for $E_8, E_7, E_6$ and $D_5$ theories in a close form expression, which are summarized in Appendix B. From the result in Appendix B, we compute the Wilson loop BPS invariants in the massless limit and check the consistency with the invariants obtained in Appendix C.3.

**BPS expansions**    Recall that we use the primitive curves $\mathsf{C}_i$ to generate the Wilson loops in the representation $\mathbf{r}_i$. For the rank-one case, there is only a single non-decomposable representation $\mathbf{r}$, so all the primitive curves $\mathsf{C}_i$ are isomorphic to each other. We will use the notation $\mathsf{n}$

to denote the n primitive curves or the representation $\mathbf{r}^{\otimes n}$. In this notation, we define the BPS sector in the representation n as

$$\mathcal{F}_{\text{BPS,n}} \equiv \mathcal{I}^{n-1} \cdot \sum_{\beta \in H_2(X, \mathbb{Z})} (-1)^{2j_L + 2j_R} \widetilde{N}^\beta_{j_L, j_R} \chi_{j_L}(\epsilon_-) \chi_{j_R}(\epsilon_+) e^{-\beta \cdot t}, \tag{37}$$

where

$$\mathcal{I} \equiv 2\sinh(\epsilon_1/2) \cdot 2\sinh(\epsilon_2/2). \tag{38}$$

Then we can obtain the BPS expansion (12) of the Wilson loop expectation values in the representation n can be simplified as

$$\langle W_{\text{n}} \rangle = \sum_{\substack{l, \text{n}_i, k_i > 0 \\ \sum_{i=1}^l \text{n}_i k_i = \text{n}}} \frac{\text{n}!}{\prod_{i=1}^l (\text{n}_i!)^{k_i} k_i!} \mathcal{F}^{k_1}_{\text{BPS,n}_1} \cdots \mathcal{F}^{k_l}_{\text{BPS,n}_l}. \tag{39}$$

## 3.2 The refined holomorphic anomaly equations for BPS sectors

In this subsection, we study the refined holomorphic anomaly equations for the BPS sectors (37) in the rank one case. The discussion here is the same as the general case in Section 2, the purpose of this subsection is to clarify the notations. Our results involve the genus expansion of the BPS sector

$$\mathcal{F}_{\text{BPS,n}} = \sum_{g=0}^\infty \sum_{n=0}^\infty (\epsilon_1 + \epsilon_2)^{2n} (\epsilon_1 \epsilon_2)^{g-1+\text{n}} \mathcal{F}^{(n,g)}_{\text{n}}. \tag{40}$$

In particular, when n = 0, we define

$$\mathcal{F}^{(n,g)}_{\text{n}=0} = \mathcal{F}^{(n,g)}, \tag{41}$$

which is the refined topological string free energy at genus $(n, g)$.

As discussed in Section 2, the holomorphic limit of the refined holomorphic anomaly equations for the Wilson loop amplitudes proposed in [34] can be rewritten in the form

$$\frac{\partial}{\partial S^{ij}} \langle W_{\mathbf{r}} \rangle = \frac{\epsilon_1 \epsilon_2}{2} \left( D_i D_j \langle W_{\mathbf{r}} \rangle + D_i \mathcal{F}_{\text{HAE}} D_j \langle W_{\mathbf{r}} \rangle + D_j \mathcal{F}_{\text{HAE}} D_i \langle W_{\mathbf{r}} \rangle \right), \tag{42}$$

which can be used to derive the refined holomorphic anomaly equation for the BPS sectors. In the rank-one case, by turning off all the mass parameters, we conclude that the BPS sector $\mathcal{F}^{(n,g)}_{\text{n}}$ satisfy the refined holomorphic anomaly equations

$$\frac{\partial \mathcal{F}^{(n,g)}_{\text{n}}}{\partial S} = \frac{1}{2} \left( D^2 \mathcal{F}^{(n,g-1)}_{\text{n}} + \sum_{n',g'}{}' \frac{\text{n}!}{\text{n}'!(\text{n}-\text{n}')!} D\mathcal{F}^{(n',g'-n)}_{\text{n}} \cdot D\mathcal{F}^{(n-n',g-g'-n')}_{\text{n}-\text{n}'} \right), \tag{43}$$

for any n > 0 and $n, g \geq 0$. Here the prime sum means we sum over all the integers $0 \leq n' \leq n$, $0 \leq g' \leq g + \text{n}$ by excluding $n' + g' = 0$ and $n' + g' = n + g + \text{n}$. Here $D$ is the covariant derivative defined as

$$D\mathcal{F}^{(n,g)}_{\text{n}} = \partial_z \mathcal{F}^{(n,g)}_{\text{n}}, \quad D^2 \mathcal{F}^{(n,g)}_{\text{n}} = (\partial_z + \Gamma)\partial_z \mathcal{F}^{(n,g)}_{\text{n}}, \tag{44}$$

$\Gamma$ is the Christoffel symbol.

In the following sections, we will use (43) to compute the BPS sectors as well as the Wilson loop BPS invariants.

| CY3 | $\mathbb{P}^2$ | $\mathbb{P}^1 \times \mathbb{P}^1$ | $D_5$ | $E_6$ | $E_7$ | $E_8$ |
|-----|------|------------|-------|-------|-------|-------|
| $c_0$ | 27 | $-16$ | 16 | 27 | 64 | 432 |
| $\kappa$ | $\frac{1}{3}$ | 1 | 4 | 3 | 2 | 1 |
| $a$ | $\frac{1}{3}$ | $\frac{1}{2}$ | 1 | 1 | 1 | 1 |

Table 1: Some constants of the local geometries.

### 3.3 Examples

This subsection gives some rank-one examples of the direct integration method for the BPS sectors of Wilson loop expectation values. We will focus on massless cases, which means we set all the mass parameters to zero. There is only one parameter left, denoted as $t$ for the Kähler parameter in the A-model or $z$ for the complex structure parameter in the B-model. The A- and B-model parameters are connected via local mirror symmetry [54]. Here we use the same notation as was used in [47]. In general, the Calabi-Yau periods $\Pi_{A,B}$ are annihilated by the Picard-Fuchs operator

$$\mathcal{L} = \Theta^3 + c_0 z \Theta \prod_{i=1}^{2}(\Theta + 1 - a_i). \tag{45}$$

For the model we consider in this paper, the values $(a_1, a_2)$ are given by

$$\mathbb{P}^2 : (\frac{1}{3}, \frac{2}{3}), \quad \mathbb{P}^1 \times \mathbb{P}^1 : (\frac{1}{2}, \frac{1}{2}), \quad D_5 : (\frac{1}{2}, \frac{1}{2}), \tag{46}$$

$$E_6 : (\frac{1}{3}, \frac{2}{3}), \quad E_7 : (\frac{1}{4}, \frac{3}{4}), \quad E_8 : (\frac{1}{6}, \frac{5}{6}), \tag{47}$$

and the constant $c_0$ are listed in Table 1. The A- and B-periods $\Pi_{A,B}$ can be solved using the Frobenius method starting with the singular term of the periods

$$\Pi_A = \log(z) + \mathcal{O}(z), \tag{48}$$

$$\Pi_B = \Pi_A^2 + \mathcal{O}(z). \tag{49}$$

The A-period $\Pi_A$ plays the role of mirror map, that maps the A-model parameter $t$ to the B-model parameter as $-t = \Pi_A$. The B-period $\Pi_B$ is proportional to the derivative of the prepotential

$$\Pi_B \propto \partial_t F^{(0,0)}, \tag{50}$$

where $F^{(0,0)} = -\frac{\kappa}{6}t^3 + \cdots$ and $\kappa$ is the triple intersection number listed in Table 1. The discriminant is $\Delta = 1 + c_0 z$. We refer the reader to [47] about other important input but less related to our later discussions.

With the notation we have introduced, the genus zero expression of the BPS sector in the fundamental representation is

$$\mathcal{F}_{\mathsf{n}=1}^{(0,0)} = \frac{1}{z^a}, \tag{51}$$

The coefficient $a$ is related to the inverse of the self-intersection number of the curve related to the Kähler parameter $t$ and the values are summarized in Table 1. Higher genus results can be solved using the direct integration method, with the holomorphic ambiguity

$$f_{\mathsf{n}}^{(n,g)}(z) = z^{-a\mathsf{n}} \left( \sum_{i=1}^{2(n+g-1)+\mathsf{n}} \frac{x_i}{\Delta^i} + \sum_{i=0}^{o} y_i z^i \right), \tag{52}$$

where $x_i, y_i$ are unknown coefficients. Here we add a factor $z^{-an} = \left(\mathcal{F}_{n=1}^{(0,0)}\right)^n$ in the holomorphic ambiguity (52) according to the expected singular behavior at the large volume limit of the Wilson loop expectation values. For the $E_8, E_7, E_6, D_5$ models, we solve the BPS sector in the fundamental representation, we fix the holomorphic ambiguities from the regularity at the conifold point and a few Wilson loop BPS invariants that are inherited from the BPS invariants of E-strings. The solvability serves as a consistency check for the holomorphic anomaly equation. We list the Wilson loop BPS invariants for those models in Section C.3.

For the local $\mathbb{P}^2$ and local $\mathbb{P}^1 \times \mathbb{P}^1$ models, the holomorphic ambiguities can be entirely fixed by the regularities of the BPS sectors at both the conifold point and the orbifold point, we will give a detailed description of the B-model calculation for them in the following sections.

### 3.3.1 local $\mathbb{P}^2$

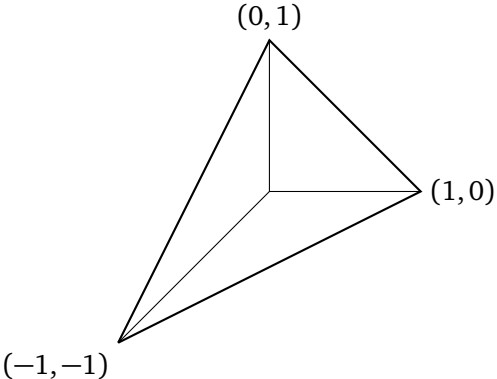

Figure 1: Toric diagram for $\mathbb{P}^2$, described by the ray vectors $(0,1), (1,0), (-1,-1)$.

The toric diagram for local $\mathbb{P}^2$ is described in Figure 1, from which we can read the Picard-Fuchs operator

$$\mathcal{L} = \Theta^3 + 3z(3\Theta + 2)(3\Theta + 1)\Theta, \tag{53}$$

where $\Theta \equiv z\frac{\partial}{\partial z}$. We can then compute the mirror map

$$-t = \log(z) - 6z + 45z^2 - 560z^3 + \frac{17325z^4}{2} - \frac{756756z^5}{5} + \mathcal{O}(z^6). \tag{54}$$

The genus one free energies are

$$\mathcal{F}^{(0,1)} = -\frac{1}{12}\log(z^7\Delta) - \frac{1}{2}\log\left|\frac{\partial t}{\partial z}\right|, \tag{55}$$

$$\mathcal{F}^{(1,0)} = \frac{1}{24}\log(z^{-1}\Delta), \tag{56}$$

where $\Delta = 1 + 27z$ is the discriminant. Then one can find the Yukawa coupling can be written in close form

$$C_{zzz} = \left(\frac{\partial t}{\partial z}\right)^3 \cdot \frac{\partial^3 \mathcal{F}^{(0,0)}}{\partial t^3} = -\frac{1}{3z^3(1 + 27z)}. \tag{57}$$

The propagator and the Christoffel symbol on the moduli space are

$$S^{zz} = \frac{2\partial_z \mathcal{F}^{(0,1)}}{C_{zzz}} = 3z^3(1+27z)\partial_z \log \frac{\partial t}{\partial z} + \frac{1}{2}z^2(7+216z), \tag{58}$$

$$\Gamma^z_{zz} = \frac{\partial z}{\partial t}\frac{\partial^2 t}{\partial z^2} = -C_{zzz}S^{zz} - \frac{7+216z}{6z(1+27z)}. \tag{59}$$

Since there is only one modulus $z$, the propagator and connection only have one component, so we will drop the indices in the symbol and use $S$ and $\Gamma$ to denote the propagator and connection.

There are two other singular points in the moduli space, which we will call the conifold point at $\Delta = 0$ and orbifold point at $\frac{1}{z} = 0$, that we can use the parameters

$$z_c = \Delta = 1 + 27z, \tag{60}$$

and

$$z_o = \frac{1}{z}, \tag{61}$$

around the conifold point and orbifold point. The Kähler parameter can then be solved from the Picard-Fuchs operator around the region when $z_c$ or $z_o$ is small, we get the mirror maps $t_c$ and $t_o$, they give the inverse expansions

$$z_c = t_c - \frac{11}{18}t_c^2 + \frac{145}{486}t_c^3 - \frac{6733}{52488}t_c^4 + \frac{120127}{2361960}t_c^5 - \frac{2431777}{127545840}t_c^6 + \mathcal{O}(t_c^7), \tag{62}$$

$$z_o = t_o^3 + \frac{1}{216}t_o^6 - \frac{1}{60480}t_o^9 + \frac{367}{1763596800}t_o^{12} - \frac{105067}{31776487142400}t_o^{15} + \mathcal{O}(t_c^{18}). \tag{63}$$

With all the ingredients, we can solve the topological string amplitudes for higher genus $n + g \geq 2$ from the refined holomorphic equation, which has been done in [47]. We will treat them as an input of the direct integration method for the BPS sector. The direct integration method involves holomorphic ambiguities, we have the ansatz

$$f_n^{(n,g)}(z) = z^{-\frac{n}{3}}\left(\sum_{i=1}^{2(n+g-1)+n}\frac{x_i}{\Delta^i} + \sum_{i=0}^{o}y_i z^i\right), \tag{64}$$

where

$$o = \left\lfloor \frac{1}{3}(2n + 2g + n)\right\rfloor. \tag{65}$$

By considering the asymptotic behavior of the amplitudes $\mathcal{F}_n^{(n,g)}$ around the conifold point, orbifold point and large volume point, the unknown coefficients $x_i, y_i$ in the ansatz (64) can be completely solved. Our first condition is that the amplitudes are regular at the conifold point

$$\mathcal{F}_n^{(n,g)} = \text{const.} + \mathcal{O}(t_c), \tag{66}$$

then all the coefficients $x_i$ can be fixed. The second condition is that the amplitudes are regular around the orbifold point

$$\mathcal{F}_n^{(n,g)} = \text{const.} + \mathcal{O}(t_o), \tag{67}$$

then all the coefficients $y_i$ with $i > \frac{n}{3}$ can be fixed. The remaining coefficient can be fixed by considering the singular behavior around the large volume point

$$\mathcal{F}^{(0,0)}_{n=1} = \frac{1}{z^{1/3}} = Q^{-\frac{1}{3}} + \mathcal{O}(Q^{\frac{2}{3}}), \tag{68}$$

and for all other genus $(n, g)$ the coefficients of $Q^{-\frac{n}{3}+\tilde{d}}$ for $-\frac{n}{3} + \tilde{d} \leq 0$ is zero. Thus, we can completely solve the BPS sectors to arbitrary genus $(n, g)$ and arbitrary representation n. Lastly, by using the refined BPS expansion with proper maximal spins, we can recover the refined BPS invariants in (37). In particular, we observe that the maximal spins have the exact form

$$j_L^{\max} = \max(0, \frac{(\tilde{d}-2)(\tilde{d}-1)}{2}), \qquad j_R^{\max} = \max(0, \frac{(\tilde{d}+1)(\tilde{d}+2)}{2} - n - 1), \tag{69}$$

where $\tilde{d} = 0, 1, 2, 3, \cdots$ is the scaled degree defined as

$$\tilde{d} = d + \frac{n}{3}. \tag{70}$$

Even though there is no limit to solving the amplitudes from the direct integration method, due to computational cost constraints, we use the direct integration method to solve the amplitudes up to $n+g+n \leq 20$, with $n \leq 12$, where the expression of the amplitudes can be found in [55]. Then we use the maximal spins to fix the refined Wilson loop BPS invariants that appear in the BPS expansion (37). For example, the refined Wilson loop BPS invariants for n $= 12$ and $d = 1$ are listed in Table 2, none of which are inherited from any known refined BPS invariants of CY3's. More refined invariants can be found in Appendix C.1.

| $2j_L \backslash 2j_R$ | 0 | 1 | 2 | 3 | 4 | 5 | 6 | 7 | 8 |
|---|---|---|---|---|---|---|---|---|---|
| 0 | 2642 | | 7176 | | 311 | | | | |
| 1 | | 1611 | | 3954 | | 79 | | | |
| 2 | 79 | | 456 | | 1379 | | 13 | | |
| 3 | | 13 | | 92 | | 377 | | 1 | |
| 4 | | | 1 | | 13 | | 79 | | |
| 5 | | | | | | 1 | | 12 | |
| 6 | | | | | | | | | 1 |

Table 2: BPS spectrum of the Wilson loop for local $\mathbb{P}^2$ with n $= 12$, $d = 1$.

### 3.3.2 local $\mathbb{P}^1 \times \mathbb{P}^1$

In this subsection, we solve the BPS sectors for local $\mathbb{P}^1 \times \mathbb{P}^1$, its toric diagram is described in Figure 2. The mirror map can be solved from the Picard-Fuchs equation, as given by

$$-t(z) = \log(z) + 4z + 18z^2 + \frac{400z^3}{3} + 1225z^4 + \frac{63504z^5}{5} + \mathcal{O}(z^6). \tag{71}$$

The genus one free energies are

$$\mathcal{F}^{(0,1)} = -\frac{1}{12} \log(z^7 \Delta) - \frac{1}{2} \log \left| \frac{\partial t}{\partial z} \right|, \tag{72}$$

$$\mathcal{F}^{(1,0)} = \frac{1}{24} \log(z^{-2} \Delta), \tag{73}$$

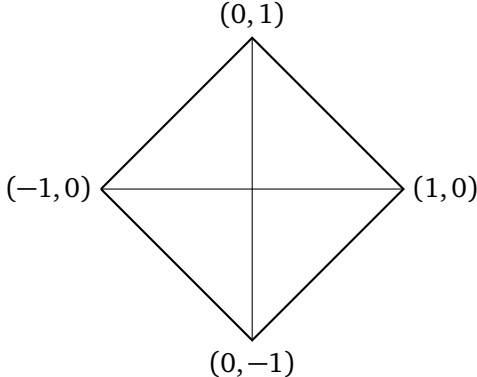

Figure 2: Toric diagram for local $\mathbb{P}^1 \times \mathbb{P}^1$, described by the ray vectors $(0,1),(1,0),(-1,0),(0,-1)$.

where $\Delta = (1-16z)$ is the discriminant of the mirror geometry. The propagator $S^{zz}$ and the Christoffel symbol $\Gamma^z_{zz}$ are defined based on the special geometry relations [21]

$$D_z S^{zz} = \partial_z S^{zz} + 2\Gamma^z_{zz} S^{zz} = -C_{zzz} S^{zz} S^{zz} - \frac{z(1-12z)}{9(1-16z)}, \tag{74}$$

$$\Gamma^z_{zz} = \frac{\partial z}{\partial t}\frac{\partial^2 t}{\partial z^2} = -C_{zzz} S^{zz} - \frac{4(1-18z)}{3z(1-16z)}. \tag{75}$$

where $C_{zzz}$ is the Yukawa coupling

$$C_{zzz} = \left(\frac{\partial t}{\partial z}\right)^3 \cdot \frac{\partial^3 \mathcal{F}^{(0,0)}}{\partial t^3} = -\frac{1}{z^3(1-16z)}, \tag{76}$$

so that the propagator is

$$S^{zz} = \frac{1}{C_{zzz}}\left(2\partial_z\mathcal{F}^{(0,1)} - \frac{1}{6z}\right) = z^3(1-16z)\partial_z \log\frac{\partial t}{\partial z} + \frac{4}{3}z^2(1-18z). \tag{77}$$

Since only one component of the propagator exists, we abbreviate $S^{zz}$ as $S$. Around the conifold point where $\Delta = 0$, we use the complex structure parameter $z_c = 1-16z$. Similarly, around the orbifold point where $z \to \infty$, we use the parameter $z_o = \frac{1}{z}$. By solving the Picard-Fuchs equation around the region when $z_c$ or $z_o$ is small, we get the mirror maps $t_c$ and $t_o$, they give the inverse expansions

$$z_c = t_c - \frac{5}{8}t_c^2 + \frac{61}{192}t_c^3 - \frac{443}{3072}t_c^4 + \frac{14993}{245760}t_c^5 - \frac{14515}{589824}t_c^6 + \mathcal{O}(t_c^7), \tag{78}$$

$$z_o = t_o^2 - \frac{1}{96}t_o^4 - \frac{11}{368640}t_o^6 - \frac{31}{41287680}t_o^8 - \frac{141941}{7610145177600}t_o^{10} + \mathcal{O}(t_o^{12}). \tag{79}$$

The BPS sectors can be calculated using the direct integration method, up to holomorphic ambiguities, in the ansatz form

$$f_n^{(n,g)}(z) = z^{-\frac{n}{2}}\left(\sum_{i=1}^{2(n+g-1)+n}\frac{x_i}{\Delta^i} + \sum_{i=0}^{\lfloor n+g+n\rfloor}y_i z^i\right). \tag{80}$$

Similar to the case of $\mathbb{P}^2$, by using the convention that around the large volume point, the only negative degree of $Q = e^{-t}$ comes from the genus zero part $\mathcal{F}_{n=1}$, specifically

$$\mathcal{F}^{(0,0)}_{n=1} = \frac{1}{\sqrt{z}}, \tag{81}$$

together with the regularity of the amplitudes at the conifold point and orbifold point, we can fix the holomorphic ambiguities for all the genera $(n, g)$ and all the representations n. In particular, we find that the maximal spins have the exact form

$$j_L^{\max} = \left\lfloor \frac{\tilde{d}}{2} \right\rfloor \left( \left\lfloor \frac{\tilde{d}}{2} \right\rfloor - 1 \right) - \frac{1}{2} \left\lfloor \frac{\tilde{d}-1}{2} \right\rfloor (1 - (-1)^{\tilde{d}}), \tag{82}$$

$$j_R^{\max} = \left\lfloor \frac{\tilde{d}}{2} \right\rfloor \left( \left\lfloor \frac{\tilde{d}}{2} \right\rfloor + 3 \right) - \frac{1}{2} \left\lfloor \frac{\tilde{d}-1}{2} \right\rfloor (1 - (-1)^{\tilde{d}}) - n + 1, \tag{83}$$

where $\tilde{d} = 0, 1, 2, 3, \cdots$, is the scaled degree defined as

$$\tilde{d} = d + \frac{n}{2}. \tag{84}$$

We use the direct integration method to solve the BPS sectors for $n + g + n \le 20$ and $n \le 10$, where the expression of the amplitudes can be found in [55]. We use the maximal spins to determine the refined Wilson loop BPS invariants. For example, the refined BPS invariants for $n = 10$ and $d = 1, 2$ are listed in Table 3 and additional invariants can be found in Appendix C.2.

| $2j_L \backslash 2j_R$ | 0 | 1 | 2 | 3 | 4 | 5 |
|---|---|---|---|---|---|---|
| 0 | | 1115 | | 10 | | |
| 1 | 89 | | 402 | | 1 | |
| 2 | | 14 | | 90 | | |
| 3 | | | 1 | | 13 | |
| 4 | | | | | | 1 |

$d = 1$

| $2j_L \backslash 2j_R$ | 0 | 1 | 2 | 3 | 4 | 5 | 6 | 7 | 8 | 9 |
|---|---|---|---|---|---|---|---|---|---|---|
| 0 | | 5192 | | 9982 | | 462 | | 2 | | |
| 1 | 594 | | 2608 | | 5792 | | 134 | | | |
| 2 | | 160 | | 754 | | 2168 | | 22 | | |
| 3 | 2 | | 24 | | 160 | | 620 | | 2 | |
| 4 | | | | 2 | | 24 | | 138 | | |
| 5 | | | | | | | 2 | | 22 | |
| 6 | | | | | | | | | | 2 |

$d = 2$

Table 3: BPS spectrum of the Wilson loop for local $\mathbb{P}^1 \times \mathbb{P}^1$ with n $= 10$ and $d \le 2$.

## 4 Magnetic dual and quantum spectrum

The magnetic dual of the topological string amplitudes can be obtained by expanding the topological string amplitudes around the conifold point [56–58]. The duality is usually called electric-magnetic duality in the gauge theory [59, 60], it maps the theory of electric particles in the weak coupling region to its strongly coupled region, which is equal to a dual theory of magnetic monopoles or monopole strings in 5D in the weakly coupling region. In this sense, the expansion parameter $t$ exchanges with the dual parameter $t_D$,

$$t_D = C \frac{\partial \mathcal{F}^{(0,0)}}{\partial t} = a' t_c, \tag{85}$$

which is proportional to the Kähler parameter $t_c$ around the conifold point. The dual parameter $t_D$ plays the role of the magnetic monopole string tension. Here the quantity $a'$ is a factor

that arises because of the notations, we have $a' = 1$ for local $\mathbb{P}^1 \times \mathbb{P}^1$ and $a' = \sqrt{3}$ for local $\mathbb{P}^2$ according to the notation of $t_c$ defined in Section 3.3.

In this section, we intend to study the magnetic dual of the Wilson loop expectation values in the NS limit $\epsilon_1 \to \hbar, \epsilon_2 \to 0$, which means we want to expand the Wilson loop expectation values around the conifold point in the NS limit. These are expected to be the expectation values of the 't Hooft operators in the magnetic dual theory. As we will see later, by imposing a proper quantization condition, the expectation values reproduce the quantum spectrum of the quantum Hamiltonians of the corresponding quantum integrable systems.

The Nekrasov-Shatashvili (NS) limit of the topological string theory is related to the integrable system [61]. In the NS limit, if the CY3 is toric, the mirror curve in the B-model is quantized as the quantum spectral curves for the cluster integrable systems [62, 63]. See [64, 65] for other related discussions on the massive quantum curves for del Pezzo surfaces. The complex structure parameter, identified with the Wilson loop expectation value in the NS limit [25, 66–69], is connected to the eigenvalue of the corresponding quantum Hamiltonian. If one treats the quantum Hamiltonian as a quantum mechanical system, the phase space of the quantum system is usually bounded, so we have a quantization condition states in [61, 70, 71] that the quantum dual parameter $t_D(\hbar)$ has the WKB quantization condition

$$t_D(\hbar) \equiv C \frac{\partial}{\partial t} \lim_{\epsilon_1 \to \hbar, \epsilon_2 \to 0} \epsilon_1 \epsilon_2 \mathcal{F} = \hbar \left( l + \frac{1}{2} \right), \quad l = 0, 1, 2, \cdots, \tag{86}$$

in terms of the energy level $l$. Denote

$$\left\langle W_D^{(n,g)} \right\rangle (t_D) \tag{87}$$

as the genus $(n, g)$ expectation values of the dual Wilson loops, expanding in terms of the dual parameter $t_D$. Classically, the eigenvalue of the Hamiltonians is equal to $\left\langle W_D^{(0,0)} \right\rangle (t_D)$, which is directly lifted to quantum version according to

$$\left\langle W_D^{(0,0)} \right\rangle (t_D) = \sum_{n=0}^{\infty} \left\langle W_D^{(n,0)} \right\rangle (t_D(\hbar)) \hbar^{2n}, \tag{88}$$

The quantum dual parameter $t_D(\hbar)$ has the expansion

$$t_D(\hbar) = t_D + \sum_{i=1}^{\infty} t_{D,2i} \hbar^{2i}, \tag{89}$$

where the quantum corrections $t_{D,2i}$ can be solved as functions of the classical dual parameter $t_D$ from the relation (88). For example, for the first two orders, we have

$$
\begin{aligned}
t_{D,2} &= -\frac{W_D^{(1)}}{W_{D,1}^{(0)}}, \\
t_{D,4} &= -\frac{W_D^{(2)}}{W_{D,1}^{(0)}} + \frac{W_D^{(1)} W_{D,1}^{(1)}}{\left( W_{D,1}^{(0)} \right)^2} - \frac{\left( W_D^{(1)} \right)^2 W_{D,2}^{(1)}}{2 \left( W_{D,1}^{(0)} \right)^3},
\end{aligned}
$$

$$\tag{90}$$

where we have used the notations

$$W_{D,k}^{(n)} \equiv \partial_{t_D}^k \left\langle W_D^{(n,0)} \right\rangle (t_D), \qquad W_D^{(n)} \equiv W_{D,0}^{(n)}. \tag{91}$$

Then the explicit values of the quantum corrections to $t_D(\hbar)$ can be solved from the expression of the Wilson loop expectation value around the conifold point. After solving the quantum corrected dual parameter $t_D(\hbar)$, one can solve the quantization condition for the classical dual parameter $t_D$ from (86), and then substitute these solutions back into $\langle W_D^{(0,0)}\rangle(t_D)$, we can then solve the energy spectrum

$$E_l = \left\langle W_D^{(0,0)}\right\rangle(t_D). \tag{92}$$

For example, for local $\mathbb{P}^1 \times \mathbb{P}^1$, the quantization of the dual parameter is

$$t_D = \hbar\left(l + \frac{1}{2}\right) + \frac{\hbar^2}{32} - \frac{2l+1}{512}\hbar^3 + \mathcal{O}(\hbar^4). \tag{93}$$

By substituting (93) into (92), we get the quantized energy spectrum

$$E_l = 4 + (2l+1)\hbar + \frac{1}{8}(2l^2 + 2l + 1)\hbar^2 + \frac{1}{192}(2l^3 + 3l^2 + 3l + 1)\hbar^3 + \mathcal{O}(\hbar^4), \tag{94}$$

which agrees with the results in [72].

Note that our method here, in principle, is no different than the method in [72]. However, the advantage here is that we do not use any information about the explicit expression of the quantum curve or the quantum Hamiltonian, we solve the quantum spectrum only from the conifold point expansion of the Wilson loop expectation value.

# 5 Blowup equations and Wilson loops

The blowup equations, which are functional equations over the partition function of refined topological strings, were initially derived for 4D instanton partition functions [38]. These equations served as generalizations of the contact term equation [73]. In [39,40], these equations were generalized to the K-theoretic version for 5D $\mathcal{N} = 1$ $SU(N)$ gauge theory, which clarify the connection between the K-theoretic instanton partition function on $\mathbb{C}^2$ and on $\widehat{\mathbb{C}}^2$. [5] Later, these blowup equations were further generalized to refined topological string theory [41], and they provide the most powerful and efficient way of computing BPS invariants and instanton partition functions for various 5D/6D quantum field theories with eight supercharges [43,76–84].

In this section, we generalize the blowup equation in a more general form, which involves the Wilson loop expectation values. We will then use these blowup equations to check the Wilson loop BPS invariants for local $\mathbb{P}^2$ and $\mathbb{P}^1 \times \mathbb{P}^1$ we calculated in 3.3.

## 5.1 Blowup equations

Define the whole topological string partition function on the Calabi-Yau threefold $X$ as

$$Z(\epsilon_1, \epsilon_2, t) = e^{\mathcal{E}} Z_{\text{BPS}}(\epsilon_1, \epsilon_2, t), \tag{95}$$

by following the notation in Section 2, where $t_i$ are the kähler parameters. Recall that the Kähler parameters are usually written as a collection of Coulomb parameters $\alpha_i$ and mass parameters $m_i$ [6] in the 5D supersymmetric gauge theories

$$t = \{\alpha_1, \cdots, \alpha_{b_4}; m_1, \cdots, m_{b_2-b_4}\}, \tag{96}$$

---

[5]See [74,75] for the developments in 4D versions with surface defects.

[6]We treat the 5D instanton counting parameter(s) as the mass parameter(s)

where $b_2$ and $b_4$ are the Betti numbers that count the number of independent compact divisors and the number of independent curves of $X$ respectively. We define $-C_{ij}$ as the intersection matrix between the compact divisors and compact curves. Then there exists a sequence of blowup equations which were first derived in [39], that connect the partition function $Z$ and the partition function $\widehat{Z}$ on the blown-up space of $\mathbb{R}^4$. They can be summarized in the form

$$\Lambda(\epsilon_1, \epsilon_2, t_i) Z(\epsilon_1, \epsilon_2, t_i + \pi i B_i) = \widehat{Z}(\epsilon_1, \epsilon_2, t_i + \pi i B_i)$$
$$= \sum_{\mathbf{n} \in \mathbb{Z}^{b_4}} (-1)^{|\mathbf{n}|} Z(\epsilon_1, \epsilon_2 - \epsilon_1, t_i + R_i \epsilon_1 + \pi i B_i)$$
$$\times Z(\epsilon_1 - \epsilon_2, \epsilon_2, t_i + R_i \epsilon_2 + \pi i B_i), \tag{97}$$

and being classified by the magnetic fluxes $R_i = n_j C_{ji} + B_i/2$. We call the function $\Lambda(\epsilon_1, \epsilon_2, t_i)$ the Lambda factor. Here we use the bold $\mathbf{n} = (n_1, \cdots, n_{b_4})$ to distinguish from the $n$ that appears in the genus expansion. Note that the magnetic fluxes for the mass parameters are constant values $R_i = \frac{1}{2} B_i, i > b_4$. The magnetic fluxes $B_i$ are always integers that satisfy the *flux quantization condition*

$$(-1)^{2j_L + 2j_R + 1} = (-1)^{\beta \cdot B}, \tag{98}$$

for any BPS particle with spin $(j_L, j_R)$ and degrees $\beta_i$.

The blowup equations (97) can be regarded as functional equations of the partition function of topological strings. It was studied in [41] by using the modularity of the B-model topological string amplitudes, the factor $\Lambda(\epsilon_1, \epsilon_2, t_i)$ is a modular function and is free of $\epsilon_1, \epsilon_2$ poles if one imposes the flux quantization condition (98). Such properties give a strong constraint on the form of $\Lambda(\epsilon_1, \epsilon_2, t_i)$, one possible solution is that $\Lambda(\epsilon_1, \epsilon_2, t_i)$ is a "constant" that is independent of the Coulomb parameters $\alpha_i$, such that there would be a bound on the possible values for the fluxes of the mass parameters, and then $\Lambda(\epsilon_1, \epsilon_2, t_i)$ can be completely determined by taking the limit $\alpha_i \to \infty$ on both side of the equation (97)

$$\Lambda(\epsilon_1, \epsilon_2, t_i) = \lim_{\alpha_i \to 0} \frac{\widehat{Z}(\epsilon_1, \epsilon_2, t_i + \pi i B_i)}{Z(\epsilon_1, \epsilon_2, t_i + \pi i B_i)} = \lim_{\alpha_i \to 0} \exp(f_i(\mathbf{n}, \epsilon_1, \epsilon_2) t_i), \tag{99}$$

by only using the classical geometric information from $\mathcal{E}$ [7], where we define $f_i(\mathbf{n}, \epsilon_1, \epsilon_2)$ from

$$f_i(\mathbf{n}, \epsilon_1, \epsilon_2) t_i$$
$$= \mathcal{E}(\epsilon_1, \epsilon_2 - \epsilon_1, t_i + R_i \epsilon_1 + \pi i B_i) + \mathcal{E}(\epsilon_1 - \epsilon_2, \epsilon_2, t_i + R_i \epsilon_2 + \pi i B_i) - \mathcal{E}(\epsilon_1, \epsilon_2, t_i + \pi i B_i). \tag{100}$$

The finiteness of the limit on the right-hand side of (99) usually indicates that the fluxes of the mass parameters are bounded even though it seems that there is no reason for the existence of this bound from the flux quantization condition. In this paper, we want to generalize the blowup equation to the cases when the magnetic fluxes of the mass parameters exceed that bound, and in this case, the Lambda factor $\Lambda(\epsilon_1, \epsilon_2, t_i)$ *does* depend on the Coulomb parameters.

## 5.2 General structure of blowup equations

The blowup equations are always solved within a bound that the Lambda factor $\Lambda(\epsilon_1, \epsilon_2, t_i)$ doesn't depend on the Coulomb parameters. However, it is possible to generalize it to arbitrary fluxes that satisfy the flux quantization condition. We have the following conjecture:

---

[7]The second equality in (99) holds if the BPS part of the partition functions is zero under the limit $\alpha_i \to 0$. This condition is indeed true for most theories, but there are indeed some special examples that we also need to consider the BPS part.

*For refined topological strings on a non-compact Calabi-Yau threefold X, for any magnetic flux B satisfying the flux quantization condition*

$$(-1)^{2j_L+2j_R+1} = (-1)^{\beta \cdot B}, \tag{101}$$

*we have the blowup equation*

$$\Lambda(\epsilon_1, \epsilon_2, t_i) Z(\epsilon_1, \epsilon_2, t_i + \pi i B_i) = \widehat{Z}(\epsilon_1, \epsilon_2, t_i + \pi i B_i)$$
$$\equiv \sum_{\mathbf{n} \in \mathbb{Z}^{b_4}} (-1)^{|\mathbf{n}|} Z(\epsilon_1, \epsilon_2 - \epsilon_1, t_i + R_i \epsilon_1 + \pi i B_i)$$
$$\times Z(\epsilon_1 - \epsilon_2, \epsilon_2, t_i + R_i \epsilon_2 + \pi i B_i), \tag{102}$$

*where the factor $\Lambda(\epsilon_1, \epsilon_2, t)$ is a linear combination of Wilson loop expectation values as*

$$\Lambda(\epsilon_1, \epsilon_2, t) = \sum_k \Lambda_k(\epsilon_1, \epsilon_2, m) \langle W_{\mathbf{r}_k} \rangle. \tag{103}$$

*Here $\Lambda_k(\epsilon_1, \epsilon_2, m)$ is a function that only depend on the Omega-deformed parameters $\epsilon_{1,2}$ and the mass parameter m. $\langle W_{\mathbf{r}_k} \rangle$ is the Wilson loop expectation value in the representation $\mathbf{r}_k$ of the gauge group. By considering the limit $t \to 0$ for the formal expression of $\Lambda$*

$$\Lambda(\epsilon_1, \epsilon_2, t_i) = \frac{\widehat{Z}(\epsilon_1, \epsilon_2, t_i + \pi i B_i)}{Z(\epsilon_1, \epsilon_2, t_i + \pi i B_i)}, \tag{104}$$

*the expression of $\Lambda(\epsilon_1, \epsilon_2, t)$ can be determined completely from the perturbative information $f_i(\mathbf{n}, \epsilon_1, \epsilon_2)$ and a few BPS invariants of the partition function and Wilson loop observables.*

Several reasons support the conjecture. The first reason is that a similar form has appeared in [40] for $SU(N)$ case, which should be generalized to arbitrary non-compact Calabi-Yau threefolds. In [40], they developed the blowup equation for the "time" dependent partition function, the expansion of the "time" variables involves the Wilson loop expectation values. The second reason is that from the formal structure of the blowup equation (102), one can verify that the factor $\Lambda(\epsilon_1, \epsilon_2, t)$ satisfies the refined holomorphic anomaly equation [85], coincides with the refined holomorphic anomaly equation (22) we have derived in Section 2 for the Wilson loop expectation values. This coincidence gives strong support for the conjecture. The third reason is that from the modular property of the topological string amplitudes, it was shown in [41] that the Lambda factor $\Lambda(\epsilon_1, \epsilon_2, t)$ is a weight zero (quasi-)modular function. In [41], the $\Lambda(\epsilon_1, \epsilon_2, t)$ factor was chosen to be a "constant" that doesn't depend on any Coulomb parameter. However, the other possible generalization would be a rational function of the complex structure parameter at the genus zero order, which is generalized to the linear combination of Wilson loop expectation values at higher genus.

**Example** To clarify our statement, we give an example. In the pure $SU(2)$ case, whose geometry is corresponding to local $\mathbb{P}^1 \times \mathbb{P}^1$, when we choose the flux of the Coulomb parameter $t$ to be $2n$, and the flux for instanton counting parameter to be $B_m$, then the perturbative contribution to the blowup equations is

$$2n(n - B_m/2)t + n^2 m + (\frac{n}{3} - \frac{4}{3}n^3 + n^2 B_m)(\epsilon_1 + \epsilon_2). \tag{105}$$

Here $B_m$ should be an even integer according to (101). When $B_m = -4, -2, 0, 2, 4$, the minimal value of $f_t = 2n(n - B_m/2)$ is always zero, from which we can derive that $\Lambda(\epsilon_1, \epsilon_2, t_i) = 1$ for $B_m = -2, 0, 2$ and

$$\Lambda(\epsilon_1, \epsilon_2, t_i) = \begin{cases} 1 - q_1 q_2, & \text{if } B_m = 4, \\ 1 - q_1^{-1} q_2^{-1}, & \text{if } B_m = -4. \end{cases} \tag{106}$$

These magnetic fluxes are those discussed in [41]. When $B_m = 6$, at $n = 1$, $f_t = -1$, which is negative, such a negative term contributes a term $e^t$ in $\Lambda(\epsilon_1, \epsilon_2, t_i)$. In the large $t$ limit, such a negative power cancel with the BPS invariants in the blowup equation contributes addition terms in $\Lambda(\epsilon_1, \epsilon_2, t_i)$. We may get

$$\Lambda(\epsilon_1, \epsilon_2, t_i)|_{t \to \infty} = 1 - Q_m(q_1 q_2(1 + q_1)(1 + q_2) + q_1^2 q_2^2 Q^{-1}) - q_1^2 q_2^2 Q_m^2, \tag{107}$$

which can be easily deduced from the perturbative prepotential and the degree one BPS invariants of $Q$. As we have explained, the $\Lambda(\epsilon_1, \epsilon_2, t_i)$ function here is pole free from $\epsilon_{1,2} \to 0$, and itself should be a modular function, among all the physical observables, the Wilson loop partition functions satisfy all the properties. Thus, we claim that the whole expression of $\Lambda$ is to replace the negative $Q$ term with its Wilson loop expectation value. For example

$$\langle W_3 \rangle = \langle W_{2 \otimes 2} \rangle - 1 = \frac{1}{Q} + 1 + 2Q_m + \mathcal{O}(Q), \tag{108}$$

then we conclude that

$$\Lambda(\epsilon_1, \epsilon_2, t_i) = 1 - Q_m \left( q_1 q_2(1 + q_1 + q_2) + q_1^2 q_2^2 \langle W_3 \rangle \right) + q_1^2 q_2^2 Q_m^2, \tag{109}$$

is the Lambda factor for the magnetic flux $B_m = 6$. By using the result of the instanton partition function, we check the blowup equation with flux $B_m = 6$ up to the six-instanton order. Similarly, for $B_m = 8$, $\Lambda(\epsilon_1, \epsilon_2, t_i)$ contains $Q_m^3$, we check the result up to the six-instanton level.

When the Coulomb parameter has half integer flux $n + \frac{1}{2}$, we also find $\Lambda(\epsilon_1, \epsilon_2, t_i)$ agrees with our prediction. We have checked

$$\Lambda(\epsilon_1, \epsilon_2, t_i)_{B_m=0} = 0,$$
$$\Lambda(\epsilon_1, \epsilon_2, t_i)_{B_m=2} = i(Q_m q_1 q_2)^{\frac{1}{4}},$$
$$\Lambda(\epsilon_1, \epsilon_2, t_i)_{B_m=4} = i(Q_m q_1^2 q_2^2)^{\frac{1}{4}} \langle W_2 \rangle,$$
$$\Lambda(\epsilon_1, \epsilon_2, t_i)_{B_m=6} = i(Q_m q_1^3 q_2^3)^{\frac{1}{4}} \left( -1 + Q_m(-1 + q_1 + q_2 + q_1 q_2) - q_1^2 q_2^2 Q_m^2 + \langle W_{2 \otimes 2} \rangle \right),$$
$$\Lambda(\epsilon_1, \epsilon_2, t_i)_{B_m=8} = \cdots. \tag{110}$$

We have checked these $\Lambda$'s numerically to the six-instanton level by using the Wilson loop expectation values obtained in [34].

**Genus zero expression**  Now we study the genus expansion of the blowup equation, we will focus on the model local $\mathbb{P}^1 \times \mathbb{P}^1$ and the leading order contribution in the blowup equations. Denote $\mathcal{F}^{(n,g)}(\epsilon_1, \epsilon_2, t, m)$ to be the genus $(n, g)$ free energy of the topological strings on local $\mathbb{P}^1 \times \mathbb{P}^1$, then the leading order expansion of the blowup equation indicates that the genus zero part of Lambda $\Lambda_{B_m, a}^{(n,g)}$ has the expression

$$\Lambda_{B_m}^{(0,0)} = \sum_{n \in \mathbb{Z} + a} \exp \left( 2n^2 \partial_t^2 \mathcal{F}^{(0,0)} + 2B_m n \partial_m \partial_t \mathcal{F}^{(0,0)} + \frac{1}{2} B_m^2 \partial_m^2 \mathcal{F}^{(0,0)} + \mathcal{F}^{(0,1)} - \mathcal{F}^{(1,0)} \right). \tag{111}$$

If our conjecture for the Lambda factor is correct, then the genus zero part is a Laurent polynomial of $z$. Indeed, as we have checked, if $a = 0$, they have the exact expressions

$$\Lambda_{B_m=0}^{(0,0)} = \Lambda_{B_m=2}^{(0,0)} = 1 \tag{112}$$

$$\Lambda_{B_m=4}^{(0,0)} = 1 - Q_m, \tag{113}$$

$$\Lambda_{B_m=6}^{(0,0)} = -\frac{Q_m}{z} + (1 - Q_m)^2, \tag{114}$$

$$\Lambda_{B_m=8}^{(0,0)} = -\frac{Q_m}{z^2} + (1 - Q_m)^4, \tag{115}$$

$$\Lambda_{B_m=10}^{(0,0)} = -\frac{Q_m}{z^3} + \frac{Q_m(2 - 3Q_m + Q_m^3)}{z^2} - \frac{3Q_m(1 - Q_m)^4}{z} + (1 - Q_m)^6, \tag{116}$$

$$\vdots$$

which are linear combinations of the genus zero Wilson loop expectation values with different representations. Interestingly, the components in the combination have the same charge under the one-form symmetry $\mathbb{Z}_2$ that maps $\sqrt{z}$ to $-\sqrt{z}$.

## 5.3   General structure of blowup equations for Wilson loops

The blowup equations can be generalized to the Wilson loop observables [11]. In the general form, we have the following conjecture:

*For refined topological strings on a non-compact Calabi-Yau threefold X, define the partition function with the insertion of the Wilson loop operator as*

$$Z_{W_{\mathbf{r}}} = \langle W_{\mathbf{r}} \rangle \cdot Z(\epsilon_1, \epsilon_2, t_i), \tag{117}$$

*where*

$$\langle W_{\mathbf{r}} \rangle = \langle W_{\mathbf{r}} \rangle(\epsilon_1, \epsilon_2, t_i) \tag{118}$$

*is the Wilson loop expectation value in the representation $\mathbf{r}$. Then for any magnetic flux B satisfying the flux quantization condition*

$$(-1)^{2j_L + 2j_R + 1} = (-1)^{\beta \cdot B}, \tag{119}$$

*and for any representations $\mathbf{r}_a, \mathbf{r}_b$, we have the blowup equation*

$$\Lambda(\epsilon_1, \epsilon_2, t_i) Z(\epsilon_1, \epsilon_2, t_i + \pi i B_i)$$
$$= \sum_{\mathbf{n} \in \mathbb{Z}^{b_4}} (-1)^{|\mathbf{n}|} Z_{W_{\mathbf{r}_a}}(\epsilon_1, \epsilon_2 - \epsilon_1, t_i + R_i \epsilon_1 + \pi i B_i)$$
$$\times Z_{W_{\mathbf{r}_b}}(\epsilon_1 - \epsilon_2, \epsilon_2, t_i + R_i \epsilon_2 + \pi i B_i), \tag{120}$$

*where the factor $\Lambda(\epsilon_1, \epsilon_2, t)$ is a linear combination of Wilson loop expectation values as*

$$\Lambda(\epsilon_1, \epsilon_2, t) = \sum_k \Lambda_k(\epsilon_1, \epsilon_2, m) \langle W_{\mathbf{r}_k} \rangle, \tag{121}$$

*Here $\Lambda_k(\epsilon_1, \epsilon_2, m)$ is a function that only depend on the Omega-deformed parameters $\epsilon_{1,2}$ and the mass parameter m. $\langle W_{\mathbf{r}_k} \rangle$ is the Wilson loop expectation value in the representation $\mathbf{r}_k$ of the gauge group.* The way to determine the explicit expression of the Lambda factor (121) is the same as the case in Section 5.2.

Note that for the BPS particles of Wilson loops, the flux quantization condition (119) also depends on the representation n as we can verify from the BPS spectra listed in Appendix C. The flux quantization condition (119) should be considered for the BPS particles of the conventional refined topological strings without the insertion of Wilson loops.

# 6 Conclusions

In this paper, we study the refined topological string correspondence of the Wilson loop operators in the five-dimensional $\mathcal{N} = 1$ supersymmetric quantum field theory on the Omega deformed background $\mathbb{R}^4_{\epsilon_1,\epsilon_2} \times S^1$. For the 5D theory which can be obtained from the M-theory compactification on the non-compact Calabi-Yau threefold $X$, the Wilson loops are provided by inserting the background non-compact primitive curves $C_1, \cdots, C_n$ on the Calabi-Yau background, and the expectation values of the Wilson loop operators can be obtained by considering the topological strings on the background $(X, \{C_1, \cdots, C_n\})$.

The expectation value of the Wilson loop operator can be written in terms of the BPS sectors, and each BPS sector has a refined BPS expansion or equivalently the refined Gopakumar-Vafa expansion that is similar to the case of refined topological strings but with an additional momentum factor. Based on the refined holomorphic anomaly equations proposed in [34], we derive the refined holomorphic anomaly equation for the BPS sectors. In particular, we use the direct integration method to compute the BPS sectors for many rank-one models, including local $E_n$ del Pezzos and local $\mathbb{P}^2$ and local $\mathbb{P}^1 \times \mathbb{P}^1$. For the last two models, we solve the BPS sectors in the B-model by using the direct integration method and recover the refined BPS invariants for them to very high representations, indicating the existence of new integral invariants.

Even though we give a general description for the Wilson loops and topological strings correspondence, all the models we have checked are toric Calabi-Yau threefolds. In the gauge theory, at least when the gauge groups are classical Lie groups, one can use the localization method to compute the Wilson expectation values. It is also interesting to study the B-model approach, particularly in non-toric cases and those cases without a gauge theory description as discussed in [86]. Some consistency checks have been done for E-strings in [31] and for the 5D rank-two cases in [87], by studying the quantum periods of the quantum curves for 5D $Sp(2)$ gauge theories. It is also interesting to verify the calculations by using the B-model method.

In Section 4, we study the Wilson loop expectation values around the conifold point in the NS limit, and without the consideration of the quantum curve, we recover the quantum spectra of the corresponding integrable systems. However, the spectrum we obtained is supposed to be the perturbative spectrum, which is consistent when the Planck constant $\hbar$ is small. Recently, the resurgence structure of the Wilson loop in the NS limit is discussed, which leads to a non-perturbative completion of the Wilson loops valid for large $\hbar$. It is interesting to see if we can obtain the non-perturbative spectrum obtained in [88, 89] from the non-perturbative Wilson loop expectation values.

In Section 5, we present generalizations of the blowup equations. We propose that when the magnetic fluxes $B_m$ for the mass parameters are large, the Lambda factor $\Lambda(\epsilon_1, \epsilon_2, t)$ in the blowup equation involves Wilson loop expectations. We give an explicit check for the case of 5D pure $SU(2)$ theory. We then generalize the formalism to the case of Wilson loops. Our proposal here can be directly generalize to 6D cases. In six dimensions, the Wilson loop becomes the Wilson surface but the expectation value of the Wilson surface operator can be effectively calculated as the expectation value of the Wilson loop operator in the 5D KK theory. Our proposal provides a generalization of the elliptic blowup equation and it is interesting to study the 6D cases to see whether new information on the elliptic genera can be obtained.

## Acknowledgements

We thank Jin Chen, Min-xin Huang, Qiang Jia, Yongchao Lü, Minsung Kim, Sung-Soo Kim, Albrecht Klemm, Kimyeong Lee, Yuji Sugimoto, Kaiwen Sun, and Longting Wu for related collaborations or discussions. XW is supported by KIAS Individual Grant QP079202.

## A    The partition function of E-string theory

In this appendix, we review the refined topological vertex formalism for the E-string theory or equivalently, the effective 5D KK theory $SU(2) + 8F$ that was studied in [53]. We start with the definitions of a sequence of functions that involve the partitions $\mu_i$:

$$\mathcal{N}_{\mu_1\mu_2}(Q;t,q) = \prod_{(i,j)\in\mu_1}\left(1 - Qt^{-\mu_{2,j}^t+i-1}q^{-\mu_{1,i}+j}\right)\cdot \prod_{(i,j)\in\mu_2}\left(1 - Qt^{\mu_{1,j}^t-i}q^{\mu_{2,i}-j+1}\right), \quad (122)$$

$$\tilde{Z}_\mu(t,q) = \prod_{(i,j)\in\mu}\left(1 - t^{\mu_j^t-i+1}q^{\mu_i-j}\right)^{-1}, \quad (123)$$

$$\mathcal{M}(Q;t,q) = \prod_{i,j=1}^{\infty}\left(1 - Qt^{i-1}q^j\right)^{-1} \quad (124)$$

$$Z_{\mu_1\mu_2} = \frac{q^{||\mu_2||^2}t^{||\mu_1^t||^2}\tilde{Z}_{\mu_1}(t,q)\tilde{Z}_{\mu_1^t}(q,t)\tilde{Z}_{\mu_2}(t,q)\tilde{Z}_{\mu_2^t}(q,t)}{\mathcal{N}_{\mu_1\mu_2}(Q^2)\mathcal{N}_{\mu_1\mu_2}(Q^2\frac{t}{q})}, \quad (125)$$

$$Z_M = \frac{\mathcal{M}(Q^2)\mathcal{M}(Q^2\frac{t}{q})}{\prod_{k=1,3,5,7}\mathcal{M}(M_kQ\sqrt{\frac{t}{q}})\mathcal{M}(\frac{M_k}{Q}\sqrt{\frac{t}{q}})}, \quad (126)$$

and

$$Z_{\mu_1\mu_2\mu_i}(M_j,M_k) = q^{\frac{||\mu_i||^2}{2}}t^{\frac{||\mu_i^t||^2}{2}}\tilde{Z}_{\mu_i}(t,q)\tilde{Z}_{\mu_i^t}(q,t)\mathcal{N}_{\mu_i\mu_1}(\frac{Q}{M_j}\sqrt{\frac{t}{q}})\mathcal{N}_{\mu_2\mu_i}(QM_j\sqrt{\frac{t}{q}})$$

$$\times \left(-\frac{M_k}{Q}\right)^{|\mu_i|}\left(\frac{1}{\sqrt{M_jM_k}}\right)^{|\mu_2|}\left(\sqrt{\frac{M_j}{M_k}}\right)^{|\mu_1|}. \quad (127)$$

Here we use the notation $q = e^{\epsilon_1}, t = e^{-\epsilon_2}$ for the Omega-deformed parameters. Then up to an extra factor $Z_{\text{extra}}^{E_8^{(1)}}$, the partition function of $SU(2) + 8F$ is

$$Z^{E_8^{(1)}}Z_{\text{extra}}^{E_8^{(1)}} = Z_M \sum_{\mu_1,\mu_2}u^{|\mu_1|+|\mu_2|}Z_{\mu_1\mu_2}\sum_{\mu_3}Z_{\mu_1\mu_2\mu_3}(M_1,M_2)\sum_{\mu_4}Z_{\mu_1\mu_2\mu_4}(M_3,M_4)$$

$$\times \sum_{\mu_5}Z_{\mu_1\mu_2\mu_5}(M_5,M_6)\sum_{\mu_6}Z_{\mu_1\mu_2\mu_6}(M_7,M_8), \quad (128)$$

where $M_k, k = 1,\cdots,8$ are the mass parameters for the eight fundamental flavors, $Q$ is the Coulomb parameter. $u$ is the instanton counting parameter for the $SU(2) + 8F$ theory and we call $|\mu_1| + |\mu_2|$ the instanton number.

In the expression for the instanton part of the E-string partition function, we also use the

notation [8]

$$\mathcal{Z}_{\mu_1\mu_2}(M_j, M_k) = \frac{\sum_{\mu_i} Z_{\mu_1\mu_2\mu_i}(M_j, M_k)}{\sum_{\mu_i} Z_{\emptyset\emptyset\mu_i}(M_j, M_k)}, \tag{129}$$

where

$$\sum_{\mu_i} Z_{\emptyset\emptyset\mu_i}(M_j, M_k) = \frac{\mathcal{M}(M_j M_k)\mathcal{M}(\frac{M_k}{M_j}\frac{t}{q})}{\mathcal{M}(\frac{M_k}{Q}\sqrt{\frac{t}{q}})\mathcal{M}(QM_k\sqrt{\frac{t}{q}})}. \tag{130}$$

In (130), the numerator of the right-hand side doesn't depend on the Coulomb parameter, thus should belong to part of the extra factor $Z_{\text{extra}}^{E_8^{(1)}}$. By computing (128) on the right-hand side to higher enough instanton numbers, the extra factor is the Coulomb-independent part and can be summarized as

$$
Z_{\text{extra}}^{E_8^{(1)}} = \mathcal{M}(M_1 M_2)\mathcal{M}(\frac{M_2}{M_1}\frac{t}{q})\mathcal{M}(M_3 M_4)\mathcal{M}(\frac{M_4}{M_3}\frac{t}{q})\mathcal{M}(M_5 M_6)\mathcal{M}(\frac{M_6}{M_5}\frac{t}{q})\mathcal{M}(M_7 M_8)\mathcal{M}(\frac{M_8}{M_7}\frac{t}{q})
$$

$$
\times \text{PE}\left(\frac{qu^2}{(1-q)(1-t)(1-u^2)}\sum_{i=1}^{4}\left(1+\frac{1}{M_{2i-1}M_{2i}}+M_{2i-1}M_{2i}\right)\right)
$$

$$
\times \text{PE}\left(\frac{tu^2}{(1-q)(1-t)(1-u^2)}\sum_{i=1}^{4}\left(1+\frac{M_{2i-1}}{M_{2i}}+\frac{M_{2i}}{M_{2i-1}}\right)\right)
$$

$$
\times \text{PE}\left(\frac{qu}{(1-q)(1-t)(1-u^2)}\prod_{i=1}^{4}\left(\frac{1}{\sqrt{M_{2i-1}M_{2i}}}+\sqrt{M_{2i-1}M_{2i}}\right)\right)
$$

$$
\times \text{PE}\left(\frac{tu}{(1-q)(1-t)(1-u^2)}\prod_{i=1}^{4}\left(\sqrt{\frac{M_{2i-1}}{M_{2i}}}+\sqrt{\frac{M_{2i}}{M_{2i-1}}}\right)\right)
$$

$$
\times \text{PE}\left(\frac{2(1+qt)u^2}{(1-q)(1-t)(1-u^2)}\right). \tag{131}
$$

## B    Wilson loop expectation values for del Pezzo surfaces

As pointed out in Section 3.1, the Wilson loops for del Pezzo surfaces can be obtained from the partition function of E-strings. This appendix derives the Wilson loop expectation values for $E_8, E_7, E_6, D_5$ del Pezzos in the fundamental representation. The exact expressions are listed in equations (143), (147), (151) and (155) respectively.

By using the following identity

$$\mathcal{N}_{\nu\lambda}(Q\sqrt{\frac{t}{q}}; t, q) = (-Q)^{|\nu|+|\lambda|} t^{\frac{1}{2}(-||\lambda^t||^2+||\nu^t||^2)} q^{\frac{1}{2}(||\lambda||^2-||\nu||^2)}\mathcal{N}_{\lambda\nu}(Q^{-1}\sqrt{\frac{t}{q}}; t, q), \tag{132}$$

the combination

$$
\mathcal{Z}_{\mu_1\mu_2\mu_i}(M_j, M_k) \equiv M_k^{\frac{1}{2}(|\mu_1|+|\mu_2|)} Z_{\mu_1\mu_2\mu_i}(M_j, M_k)
$$

$$
= t^{||\mu_i^t||^2-\frac{1}{2}||\mu_1^t||^2} q^{\frac{1}{2}||\mu_1||^2}\tilde{Z}_{\mu_i}(t, q)\tilde{Z}_{\mu_i^t}(q, t)\mathcal{N}_{\mu_1\mu_i}(\frac{M_j}{Q}\sqrt{\frac{t}{q}})\mathcal{N}_{\mu_2\mu_i}(QM_j\sqrt{\frac{t}{q}})
$$

$$
\times (-Q)^{|\mu_1|}\left(\frac{M_k}{M_j}\right)^{|\mu_i|} M_j^{\frac{1}{2}(|\mu_1|+|\mu_2|)} \tag{133}
$$

---

[8]The calculation of $\mathcal{Z}_{\mu_1\mu_2}(M_j, M_k)$ is time-consuming. Since this function can be commonly used for other topological vertex calculations, we provide the results for $\mathcal{Z}_{\mu_1\mu_2}(M_j, M_k)$, with $|\mu_1|+|\mu_2| \leq 10$, in [55].

always has positive degrees of $M_k$. Let's define $Q_b = \frac{u}{\sqrt{M_2 M_4 M_6 M_8}}$, the partition function of E-strings becomes

$$
Z^{E_8^{(1)}} Z^{E_8^{(1)}}_{\text{extra}} = Z_M \sum_{\mu_1,\mu_2} Q_b^{|\mu_1|+|\mu_2|} Z_{\mu_1\mu_2} \sum_{\mu_3} \mathcal{Z}_{\mu_1\mu_2\mu_3}(M_1, M_2) \sum_{\mu_4} \mathcal{Z}_{\mu_1\mu_2\mu_4}(M_3, M_4)
$$
$$
\times \sum_{\mu_5} \mathcal{Z}_{\mu_1\mu_2\mu_5}(M_5, M_6) \sum_{\mu_6} \mathcal{Z}_{\mu_1\mu_2\mu_6}(M_7, M_8), \tag{134}
$$

and the extra term becomes

$$
Z^{E_8^{(1)}}_{\text{extra}} = \mathcal{M}(M_1 M_2)\mathcal{M}(\frac{M_2}{M_1}\frac{t}{q})\mathcal{M}(M_3 M_4)\mathcal{M}(\frac{M_4}{M_3}\frac{t}{q})
$$
$$
\mathcal{M}(M_5 M_6)\mathcal{M}(\frac{M_6}{M_5}\frac{t}{q})\mathcal{M}(M_7 M_8)\mathcal{M}(\frac{M_8}{M_7}\frac{t}{q})
$$
$$
\text{PE}\left(\frac{qQ_b^2 M_2 M_4 M_6 M_8}{(1-q)(1-t)(1-Q_b^2 M_2 M_4 M_6 M_8)} \sum_{i=1}^{4}\left(1 + \frac{1}{M_{2i-1}M_{2i}} + M_{2i-1}M_{2i}\right)\right)
$$
$$
\text{PE}\left(\frac{tQ_b^2 M_2 M_4 M_6 M_8}{(1-q)(1-t)(1-Q_b^2 M_2 M_4 M_6 M_8)} \sum_{i=1}^{4}\left(1 + \frac{M_{2i-1}}{M_{2i}} + \frac{M_{2i}}{M_{2i-1}}\right)\right)
$$
$$
\text{PE}\left(\frac{qQ_b}{(1-q)(1-t)(1-Q_b^2 M_2 M_4 M_6 M_8)} \prod_{i=1}^{4}\left(\frac{1}{\sqrt{M_{2i-1}}} + \sqrt{M_{2i-1}}M_{2i}\right)\right)
$$
$$
\text{PE}\left(\frac{tQ_b}{(1-q)(1-t)(1-Q_b^2 M_2 M_4 M_6 M_8)} \prod_{i=1}^{4}\left(\sqrt{M_{2i-1}} + \frac{M_{2i}}{\sqrt{M_{2i-1}}}\right)\right)
$$
$$
\text{PE}\left(\frac{2(1+qt)Q_b^2 M_2 M_4 M_6 M_8}{(1-q)(1-t)(1-Q_b^2 M_2 M_4 M_6 M_8)}\right), \tag{135}
$$

which always has positive degrees of $M_2, M_4, M_6, M_8$. Then, by recursively picking up the coefficients of $M_8, M_6, M_4, M_2$ one by one, we obtain the Wilson loops for $SU(2)$ with $N_f = 7, 6, 5, 4$, or equivalently $E_8, E_7, E_6, D_5$ del Pezzos.

In practice, note that

$$
\mathcal{N}_{\mu_1 \emptyset}(Q; t, q) = \prod_{(i,j)\in\mu_1}\left(1 - Qt^{i-1}q^{-\mu_{1,i}+j}\right), \tag{136}
$$

and

$$
\mathcal{N}_{\mu_1 \square}(Q; t, q) = \mathcal{N}_{\mu_1 \emptyset}(Q; t, q) \prod_{i=1}^{l(\mu_1)} \frac{\left(1 - Qt^{i-2}q^{-\mu_{1,i}+1}\right)}{\left(1 - Qt^{i-1}q^{-\mu_{1,i}+1}\right)}\left(1 - Qt^{l(\mu_1)-1}q\right). \tag{137}
$$

We define

$$
\widetilde{\mathcal{N}}_{\mu_1}(Q; t, q) = \prod_{i=1}^{l(\mu_1)} \frac{\left(1 - Qt^{i-2}q^{-\mu_{1,i}+1}\right)}{\left(1 - Qt^{i-1}q^{-\mu_{1,i}+1}\right)}\left(1 - Qt^{l(\mu_1)-1}q\right), \tag{138}
$$

so that we have

$$
\frac{Z_{\mu_1\mu_2\square}(M_j, M_k)}{Z_{\mu_1\mu_2\emptyset(M_j,M_k)}} = \frac{\sqrt{tq}}{(1-t)(1-q)}\widetilde{\mathcal{N}}_{\mu_1}(\frac{M_j}{Q}\sqrt{\frac{t}{q}}; t, q)\widetilde{\mathcal{N}}_{\mu_2}(QM_j\sqrt{\frac{t}{q}}; t, q)\frac{M_k}{M_j}\sqrt{\frac{t}{q}}. \tag{139}
$$

Using the above components, we derive the partition functions and the Wilson loop expectation values for $E_n$ del Pezzo's in the fundamental representation. For $E_8$ del Pezzo surface:

$$Z^{E_8} Z^{E_8}_{\text{extra}} = Z_{\text{M}} \sum_{\mu_1,\mu_2} Q_b^{|\mu_1|+|\mu_2|} Z_{\mu_1\mu_2} \sum_{\mu_3} \mathcal{Z}_{\mu_1\mu_2\mu_3}(M_1,M_2) \sum_{\mu_4} \mathcal{Z}_{\mu_1\mu_2\mu_4}(M_3,M_4)$$
$$\times \sum_{\mu_5} \mathcal{Z}_{\mu_1\mu_2\mu_5}(M_5,M_6) \mathcal{Z}_{\mu_1\mu_2\emptyset}(M_7,0), \tag{140}$$

$$Z^{E_8}_{W_{\text{F}}} Z^{E_8}_{\text{extra}} = Z_{\text{M}} \sum_{\mu_1,\mu_2} Z_{\mu_1\mu_2} \sum_{\mu_3} \mathcal{Z}_{\mu_1\mu_2\mu_3}(M_1,M_2) \sum_{\mu_4} \mathcal{Z}_{\mu_1\mu_2\mu_4}(M_3,M_4)$$
$$\times \sum_{\mu_5} \mathcal{Z}_{\mu_1\mu_2\mu_5}(M_5,M_6) \mathcal{Z}_{\mu_1\mu_2\emptyset}(M_7,0)$$
$$\times \left( \widetilde{\mathcal{N}}_{\mu_1}(\frac{M_7}{Q}\sqrt{\frac{t}{q}}; t,q) \widetilde{\mathcal{N}}_{\mu_2}(QM_7\sqrt{\frac{t}{q}}; t,q) \frac{1}{M_7}\sqrt{\frac{t}{q}} - C_8 \right), \tag{141}$$

where $Z^{E_8}_{\text{extra}}$ and $C_8$ are defined from the Fourier expansion of $Z^{E_8^{(1)}}_{\text{extra}}$

$$Z^{E_8^{(1)}}_{\text{extra}} = Z^{E_8}_{\text{extra}}(1 + C_8 M_8 + \mathcal{O}(M_8^2)). \tag{142}$$

Then the expectation value of the Wilson loop operator for $E_8$ del Pezzo is

$$\left\langle W_{\text{F}}^{E_8} \right\rangle = \frac{Z^{E_8}_{W_{\text{F}}}}{Z^{E_8}}, \tag{143}$$

where we use the subscript F to denote the fundamental representation. Subsequently, for $E_7$ del Pezzo surface, we have

$$Z^{E_7} Z^{E_7}_{\text{extra}} = Z_{\text{M}} \sum_{\mu_1,\mu_2} Z_{\mu_1\mu_2} \sum_{\mu_3} \mathcal{Z}_{\mu_1\mu_2\mu_3}(M_1,M_2) \sum_{\mu_4} \mathcal{Z}_{\mu_1\mu_2\mu_4}(M_3,M_4)$$
$$\times \mathcal{Z}_{\mu_1\mu_2\emptyset}(M_5,0) \mathcal{Z}_{\mu_1\mu_2\emptyset}(M_7,0), \tag{144}$$

$$Z^{E_7}_{W_{\text{F}}} Z^{E_7}_{\text{extra}} = Z_{\text{M}} \sum_{\mu_1,\mu_2} Z_{\mu_1\mu_2} \sum_{\mu_3} \mathcal{Z}_{\mu_1\mu_2\mu_3}(M_1,M_2) \sum_{\mu_4} \mathcal{Z}_{\mu_1\mu_2\mu_4}(M_3,M_4)$$
$$\times \mathcal{Z}_{\mu_1\mu_2\emptyset}(M_5,0) \mathcal{Z}_{\mu_1\mu_2\emptyset}(M_7,0)$$
$$\times \left( \widetilde{\mathcal{N}}_{\mu_1}(\frac{M_5}{Q}\sqrt{\frac{t}{q}}; t,q) \widetilde{\mathcal{N}}_{\mu_2}(QM_5\sqrt{\frac{t}{q}}; t,q) \frac{1}{M_5}\sqrt{\frac{t}{q}} - C_7 \right), \tag{145}$$

where $Z^{E_7}_{\text{extra}}$ and $C_7$ are defined from the Fourier expansion of $Z^{E_8}_{\text{extra}}$

$$Z^{E_8}_{\text{extra}} = Z^{E_7}_{\text{extra}}(1 + C_7 M_6 + \mathcal{O}(M_6^2)). \tag{146}$$

Then the expectation value of the Wilson loop operator for $E_7$ del Pezzo is

$$\left\langle W_{\text{F}}^{E_7} \right\rangle = \frac{Z^{E_7}_{W_{\text{F}}}}{Z^{E_7}}. \tag{147}$$

For $E_6$ del Pezzo surface:

$$Z^{E_6} Z^{E_6}_{\text{extra}} = Z_{\text{M}} \sum_{\mu_1,\mu_2} Z_{\mu_1\mu_2} \sum_{\mu_3} \mathcal{Z}_{\mu_1\mu_2\mu_3}(M_1,M_2) \mathcal{Z}_{\mu_1\mu_2\emptyset}(M_3,0) \mathcal{Z}_{\mu_1\mu_2\emptyset}(M_5,0) \mathcal{Z}_{\mu_1\mu_2\emptyset}(M_7,0),$$
$$\tag{148}$$

$$Z_{W_{\mathsf{F}}}^{E_6} Z_{\text{extra}}^{E_6} = Z_{\mathrm{M}} \sum_{\mu_1,\mu_2} Z_{\mu_1\mu_2} \sum_{\mu_3} \mathcal{Z}_{\mu_1\mu_2\mu_3}(M_1, M_2) \mathcal{Z}_{\mu_1\mu_2\emptyset}(M_3, 0) \mathcal{Z}_{\mu_1\mu_2\emptyset}(M_5, 0) \mathcal{Z}_{\mu_1\mu_2\emptyset}(M_7, 0)$$

$$\times \left( \widetilde{\mathcal{N}}_{\mu_1}(\frac{M_3}{Q}\sqrt{\frac{t}{q}}; t, q) \widetilde{\mathcal{N}}_{\mu_2}(QM_3\sqrt{\frac{t}{q}}; t, q) \frac{1}{M_3}\sqrt{\frac{t}{q}} - C_6 \right), \tag{149}$$

where $Z_{\text{extra}}^{E_6}$ and $C_6$ are defined from the Fourier expansion of $Z_{\text{extra}}^{E_7}$

$$Z_{\text{extra}}^{E_7} = Z_{\text{extra}}^{E_6}(1 + C_6 M_4 + \mathcal{O}(M_4^2)). \tag{150}$$

Then the expectation value of the Wilson loop operator for $E_6$ del Pezzo is

$$\left\langle W_{\mathsf{F}}^{E_6} \right\rangle = \frac{Z_{W_{\mathsf{F}}}^{E_6}}{Z^{E_6}}. \tag{151}$$

For $D_5$ del Pezzo surface:

$$Z^{D_5} Z_{\text{extra}}^{D_5} = Z_{\mathrm{M}} \sum_{\mu_1,\mu_2} Z_{\mu_1\mu_2} \mathcal{Z}_{\mu_1\mu_2\emptyset}(M_1, 0) \mathcal{Z}_{\mu_1\mu_2\emptyset}(M_3, 0) \mathcal{Z}_{\mu_1\mu_2\emptyset}(M_5, 0) \mathcal{Z}_{\mu_1\mu_2\emptyset}(M_7, 0), \tag{152}$$

$$Z_{W_{\mathsf{F}}}^{E_6} Z_{\text{extra}}^{E_6} = Z_{\mathrm{M}} \sum_{\mu_1,\mu_2} Z_{\mu_1\mu_2} \mathcal{Z}_{\mu_1\mu_2\emptyset}(M_1, 0) \mathcal{Z}_{\mu_1\mu_2\emptyset}(M_3, 0) \mathcal{Z}_{\mu_1\mu_2\emptyset}(M_5, 0) \mathcal{Z}_{\mu_1\mu_2\emptyset}(M_7, 0)$$

$$\times \left( \widetilde{\mathcal{N}}_{\mu_1}(\frac{M_1}{Q}\sqrt{\frac{t}{q}}; t, q) \widetilde{\mathcal{N}}_{\mu_2}(QM_1\sqrt{\frac{t}{q}}; t, q) \frac{1}{M_1}\sqrt{\frac{t}{q}} - C_5 \right), \tag{153}$$

where $Z_{\text{extra}}^{D_5}$ and $C_5$ are defined from the Fourier expansion of $Z_{\text{extra}}^{E_6}$

$$Z_{\text{extra}}^{E_6} = Z_{\text{extra}}^{D_5}(1 + C_5 M_2 + \mathcal{O}(M_2^2)). \tag{154}$$

Then the expectation value of the Wilson loop operator for $D_5$ del Pezzo is

$$\left\langle W_{\mathsf{F}}^{D_5} \right\rangle = \frac{Z_{W_{\mathsf{F}}}^{D_5}}{Z^{D_5}}. \tag{155}$$

## C    Refined BPS invariants

In this appendix, we present the refined Wilson loop BPS invariants, denoted $\widetilde{N}_{j_L,j_R}^d$, in the BPS sector

$$\mathcal{F}_{\text{BPS,n}} = \mathcal{I}^{\mathsf{n}-1} \cdot \sum_{d \in \mathbb{Z}_0^+ - a\mathsf{n}}^{\infty} (-1)^{2j_L + 2j_R} \widetilde{N}_{j_L,j_R}^d \chi_{j_L}(\epsilon_-) \chi_{j_R}(\epsilon_+) e^{-dt}, \tag{156}$$

where $\mathbb{Z}_0^+$ denotes the set of non-negative integers, and the constant $a$ is defined in Table 1. These invariants are relevant for local $\mathbb{P}^2$, local $\mathbb{P}^1 \times \mathbb{P}^1$, and local $D_5, E_6, E_7, E_8$ del Pezzo surfaces, as described in Appendix C.1, Appendix C.2, and Appendix C.3 respectively.

### C.1    Refined BPS invariants for local $\mathbb{P}^2$

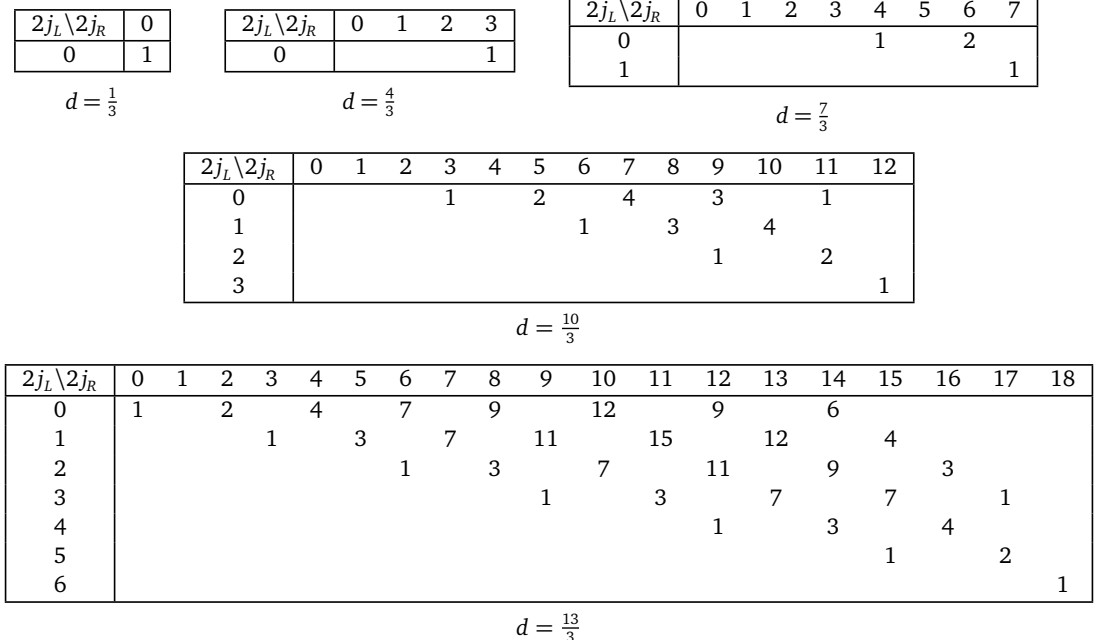

Table 4: BPS spectrum of the Wilson loop for local $\mathbb{P}^2$ with $n = 2$ and $d \leq \frac{13}{3}$.

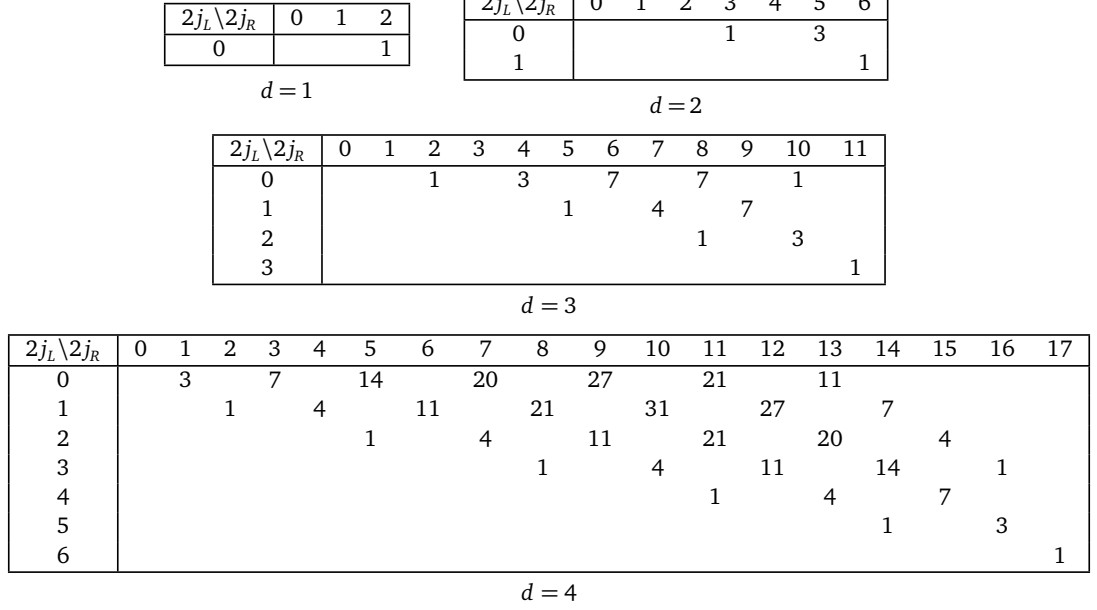

Table 5: BPS spectrum of the Wilson loop for local $\mathbb{P}^2$ with $n = 3$ and $d \leq 4$.

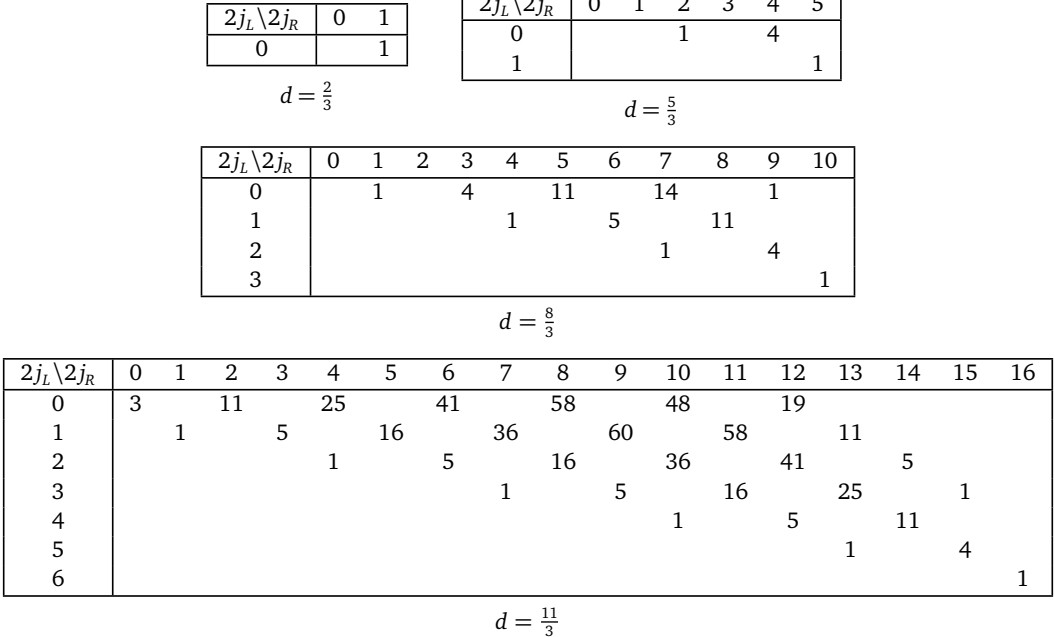

$d = \frac{2}{3}$

| $2j_L\backslash 2j_R$ | 0 | 1 |
|---|---|---|
| 0 | | 1 |

$d = \frac{5}{3}$

| $2j_L\backslash 2j_R$ | 0 | 1 | 2 | 3 | 4 | 5 |
|---|---|---|---|---|---|---|
| 0 | | | 1 | | 4 | |
| 1 | | | | | | 1 |

$d = \frac{8}{3}$

| $2j_L\backslash 2j_R$ | 0 | 1 | 2 | 3 | 4 | 5 | 6 | 7 | 8 | 9 | 10 |
|---|---|---|---|---|---|---|---|---|---|---|---|
| 0 | | 1 | | 4 | | 11 | | 14 | | 1 | |
| 1 | | | | | 1 | | 5 | | 11 | | |
| 2 | | | | | | | | 1 | | 4 | |
| 3 | | | | | | | | | | | 1 |

$d = \frac{11}{3}$

| $2j_L\backslash 2j_R$ | 0 | 1 | 2 | 3 | 4 | 5 | 6 | 7 | 8 | 9 | 10 | 11 | 12 | 13 | 14 | 15 | 16 |
|---|---|---|---|---|---|---|---|---|---|---|---|---|---|---|---|---|---|
| 0 | 3 | | 11 | | 25 | | 41 | | 58 | | 48 | | 19 | | | | |
| 1 | | 1 | | 5 | | 16 | | 36 | | 60 | | 58 | | 11 | | | |
| 2 | | | | | 1 | | 5 | | 16 | | 36 | | 41 | | 5 | | |
| 3 | | | | | | | | 1 | | 5 | | 16 | | 25 | | 1 | |
| 4 | | | | | | | | | | | 1 | | 5 | | 11 | | |
| 5 | | | | | | | | | | | | | | 1 | | 4 | |
| 6 | | | | | | | | | | | | | | | | | 1 |

Table 6: BPS spectrum of the Wilson loop for local $\mathbb{P}^2$ with $\mathsf{n} = 4$ and $d \leq \frac{11}{3}$.

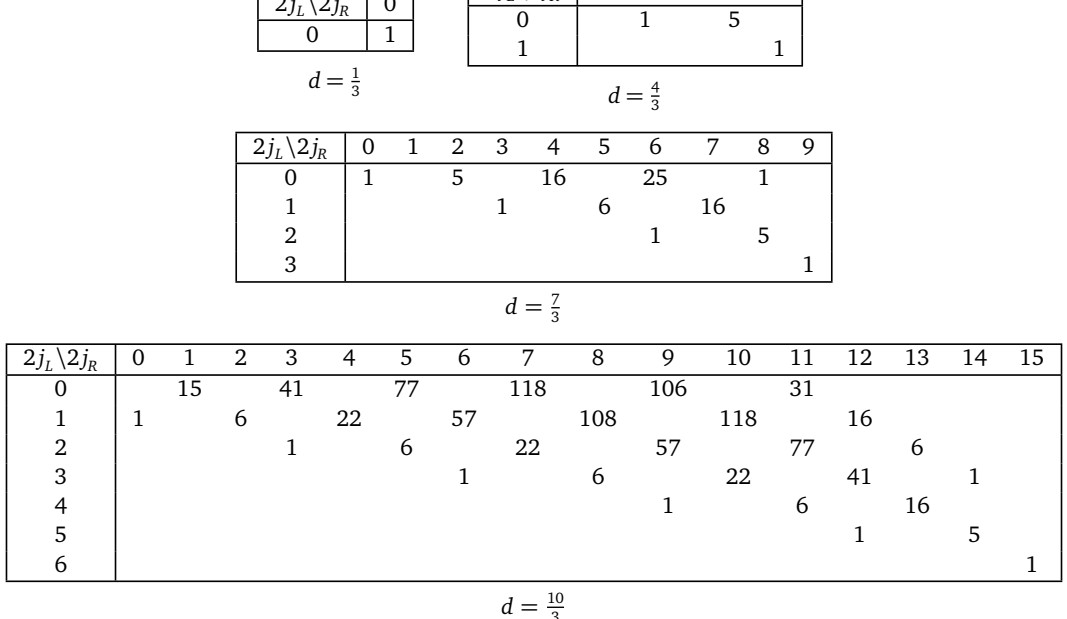

$d = \frac{1}{3}$

| $2j_L\backslash 2j_R$ | 0 |
|---|---|
| 0 | 1 |

$d = \frac{4}{3}$

| $2j_L\backslash 2j_R$ | 0 | 1 | 2 | 3 | 4 |
|---|---|---|---|---|---|
| 0 | | 1 | | 5 | |
| 1 | | | | | 1 |

$d = \frac{7}{3}$

| $2j_L\backslash 2j_R$ | 0 | 1 | 2 | 3 | 4 | 5 | 6 | 7 | 8 | 9 |
|---|---|---|---|---|---|---|---|---|---|---|
| 0 | 1 | | 5 | | 16 | | 25 | | 1 | |
| 1 | | | | 1 | | 6 | | 16 | | |
| 2 | | | | | | | 1 | | 5 | |
| 3 | | | | | | | | | | 1 |

$d = \frac{10}{3}$

| $2j_L\backslash 2j_R$ | 0 | 1 | 2 | 3 | 4 | 5 | 6 | 7 | 8 | 9 | 10 | 11 | 12 | 13 | 14 | 15 |
|---|---|---|---|---|---|---|---|---|---|---|---|---|---|---|---|---|
| 0 | | 15 | | 41 | | 77 | | 118 | | 106 | | 31 | | | | |
| 1 | 1 | | 6 | | 22 | | 57 | | 108 | | 118 | | 16 | | | |
| 2 | | | | 1 | | 6 | | 22 | | 57 | | 77 | | 6 | | |
| 3 | | | | | | | 1 | | 6 | | 22 | | 41 | | 1 | |
| 4 | | | | | | | | | | 1 | | 6 | | 16 | | |
| 5 | | | | | | | | | | | | | 1 | | 5 | |
| 6 | | | | | | | | | | | | | | | | 1 |

Table 7: BPS spectrum of the Wilson loop for local $\mathbb{P}^2$ with $\mathsf{n} = 5$ and $d \leq \frac{10}{3}$.

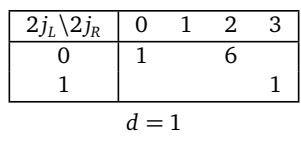

| $2j_L\backslash 2j_R$ | 0 | 1 | 2 | 3 |
|---|---|---|---|---|
| 0 | 1 | | 6 | |
| 1 | | | | 1 |

$d = 1$

| $2j_L\backslash 2j_R$ | 0 | 1 | 2 | 3 | 4 | 5 | 6 | 7 | 8 |
|---|---|---|---|---|---|---|---|---|---|
| 0 | | 6 | | 22 | | 41 | | 1 | |
| 1 | | | 1 | | 7 | | 22 | | |
| 2 | | | | | | 1 | | 6 | |
| 3 | | | | | | | | | 1 |

$d = 2$

| $2j_L\backslash 2j_R$ | 0 | 1 | 2 | 3 | 4 | 5 | 6 | 7 | 8 | 9 | 10 | 11 | 12 | 13 | 14 |
|---|---|---|---|---|---|---|---|---|---|---|---|---|---|---|---|
| 0 | 16 | | 62 | | 134 | | 226 | | 224 | | 48 | | | | |
| 1 | | 7 | | 29 | | 85 | | 182 | | 226 | | 22 | | | |
| 2 | | | 1 | | 7 | | 29 | | 85 | | 134 | | 7 | | |
| 3 | | | | | | 1 | | 7 | | 29 | | 63 | | 1 | |
| 4 | | | | | | | | | 1 | | 7 | | 22 | | |
| 5 | | | | | | | | | | | | 1 | | 6 | |
| 6 | | | | | | | | | | | | | | | 1 |

$d = 3$

Table 8: BPS spectrum of the Wilson loop for local $\mathbb{P}^2$ with $\mathsf{n} = 6$ and $d \leq 3$.

| $2j_L\backslash 2j_R$ | 0 | 1 | 2 |
|---|---|---|---|
| 0 | | 7 | |
| 1 | | | 1 |

$d = \frac{2}{3}$

| $2j_L\backslash 2j_R$ | 0 | 1 | 2 | 3 | 4 | 5 | 6 | 7 |
|---|---|---|---|---|---|---|---|---|
| 0 | 6 | | 29 | | 63 | | 1 | |
| 1 | | 1 | | 8 | | 29 | | |
| 2 | | | | | 1 | | 7 | |
| 3 | | | | | | | | 1 |

$d = \frac{5}{3}$

| $2j_L\backslash 2j_R$ | 0 | 1 | 2 | 3 | 4 | 5 | 6 | 7 | 8 | 9 | 10 | 11 | 12 | 13 |
|---|---|---|---|---|---|---|---|---|---|---|---|---|---|---|
| 0 | | 85 | | 218 | | 408 | | 450 | | 71 | | | | |
| 1 | 7 | | 37 | | 121 | | 290 | | 408 | | 29 | | | |
| 2 | | 1 | | 8 | | 37 | | 121 | | 219 | | 8 | | |
| 3 | | | | | 1 | | 8 | | 37 | | 92 | | 1 | |
| 4 | | | | | | | | 1 | | 8 | | 29 | | |
| 5 | | | | | | | | | | | 1 | | 7 | |
| 6 | | | | | | | | | | | | | | 1 |

$d = \frac{8}{3}$

Table 9: BPS spectrum of the Wilson loop for local $\mathbb{P}^2$ with $\mathsf{n} = 7$ and $d \leq \frac{8}{3}$.

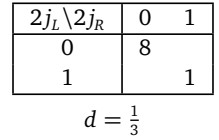

| $2j_L\backslash 2j_R$ | 0 | 1 |
|---|---|---|
| 0 | 8 | |
| 1 | | 1 |

$$d = \tfrac{1}{3}$$

| $2j_L\backslash 2j_R$ | 0 | 1 | 2 | 3 | 4 | 5 | 6 |
|---|---|---|---|---|---|---|---|
| 0 | | 36 | | 92 | | 1 | |
| 1 | 1 | | 9 | | 37 | | |
| 2 | | | | 1 | | 8 | |
| 3 | | | | | | | 1 |

$$d = \tfrac{4}{3}$$

| $2j_L\backslash 2j_R$ | 0 | 1 | 2 | 3 | 4 | 5 | 6 | 7 | 8 | 9 | 10 | 11 | 12 |
|---|---|---|---|---|---|---|---|---|---|---|---|---|---|
| 0 | 92 | | 332 | | 697 | | 858 | | 101 | | | | |
| 1 | | 45 | | 166 | | 441 | | 698 | | 37 | | | |
| 2 | 1 | | 9 | | 46 | | 166 | | 340 | | 9 | | |
| 3 | | | | 1 | | 9 | | 46 | | 129 | | 1 | |
| 4 | | | | | | | 1 | | 9 | | 37 | | |
| 5 | | | | | | | | | | 1 | | 8 | |
| 6 | | | | | | | | | | | | | 1 |

$$d = \tfrac{7}{3}$$

Table 10: BPS spectrum of the Wilson loop for local $\mathbb{P}^2$ with $\mathsf{n} = 8$ and $d \leq \tfrac{7}{3}$.

| $2j_L\backslash 2j_R$ | 0 | 1 | 2 | 3 | 4 | 5 |
|---|---|---|---|---|---|---|
| 0 | 37 | | 129 | | 1 | |
| 1 | | 10 | | 46 | | |
| 2 | | | 1 | | 9 | |
| 3 | | | | | | 1 |

$$d = 1$$

| $2j_L\backslash 2j_R$ | 0 | 1 | 2 | 3 | 4 | 5 | 6 | 7 | 8 | 9 | 10 | 11 |
|---|---|---|---|---|---|---|---|---|---|---|---|---|
| 0 | | 460 | | 1130 | | 1556 | | 139 | | | | |
| 1 | 46 | | 220 | | 645 | | 1139 | | 46 | | | |
| 2 | | 10 | | 56 | | 221 | | 506 | | 10 | | |
| 3 | | | 1 | | 10 | | 56 | | 175 | | 1 | |
| 4 | | | | | | 1 | | 10 | | 46 | | |
| 5 | | | | | | | | | 1 | | 9 | |
| 6 | | | | | | | | | | | | 1 |

$$d = 2$$

Table 11: BPS spectrum of the Wilson loop for local $\mathbb{P}^2$ with $\mathsf{n} = 9$ and $d \leq 2$.

| $2j_L \backslash 2j_R$ | 0 | 1 | 2 | 3 | 4 |
|---|---|---|---|---|---|
| 0 | | 175 | | 1 | |
| 1 | 10 | | 56 | | |
| 2 | | 1 | | 10 | |
| 3 | | | | | 1 |

$$d = \tfrac{2}{3}$$

| $2j_L \backslash 2j_R$ | 0 | 1 | 2 | 3 | 4 | 5 | 6 | 7 | 8 | 9 | 10 |
|---|---|---|---|---|---|---|---|---|---|---|---|
| 0 | 496 | | 1729 | | 2695 | | 186 | | | | |
| 1 | | 276 | | 912 | | 1784 | | 56 | | | |
| 2 | 10 | | 67 | | 287 | | 727 | | 11 | | |
| 3 | | 1 | | 11 | | 67 | | 231 | | 1 | |
| 4 | | | | | 1 | | 11 | | 56 | | |
| 5 | | | | | | | | 1 | | 10 | |
| 6 | | | | | | | | | | | 1 |

$$d = \tfrac{5}{3}$$

Table 12: BPS spectrum of the Wilson loop for local $\mathbb{P}^2$ with $\mathsf{n} = 10$ and $d \leq \tfrac{5}{3}$.

| $2j_L \backslash 2j_R$ | 0 | 1 | 2 | 3 |
|---|---|---|---|---|
| 0 | 231 | | 1 | |
| 1 | | 67 | | |
| 2 | 1 | | 11 | |
| 3 | | | | 1 |

$$d = \tfrac{1}{3}$$

| $2j_L \backslash 2j_R$ | 0 | 1 | 2 | 3 | 4 | 5 | 6 | 7 | 8 | 9 |
|---|---|---|---|---|---|---|---|---|---|---|
| 0 | | 2410 | | 4479 | | 243 | | | | |
| 1 | 287 | | 1245 | | 2697 | | 67 | | | |
| 2 | | 78 | | 365 | | 1014 | | 12 | | |
| 3 | 1 | | 12 | | 79 | | 298 | | 1 | |
| 4 | | | | 1 | | 12 | | 67 | | |
| 5 | | | | | | | 1 | | 11 | |
| 6 | | | | | | | | | | 1 |

$$d = \tfrac{4}{3}$$

Table 13: BPS spectrum of the Wilson loop for local $\mathbb{P}^2$ with $\mathsf{n} = 11$ and $d \leq \tfrac{4}{3}$.

## C.2 Refined BPS invariants for local $\mathbb{P}^1 \times \mathbb{P}^1$

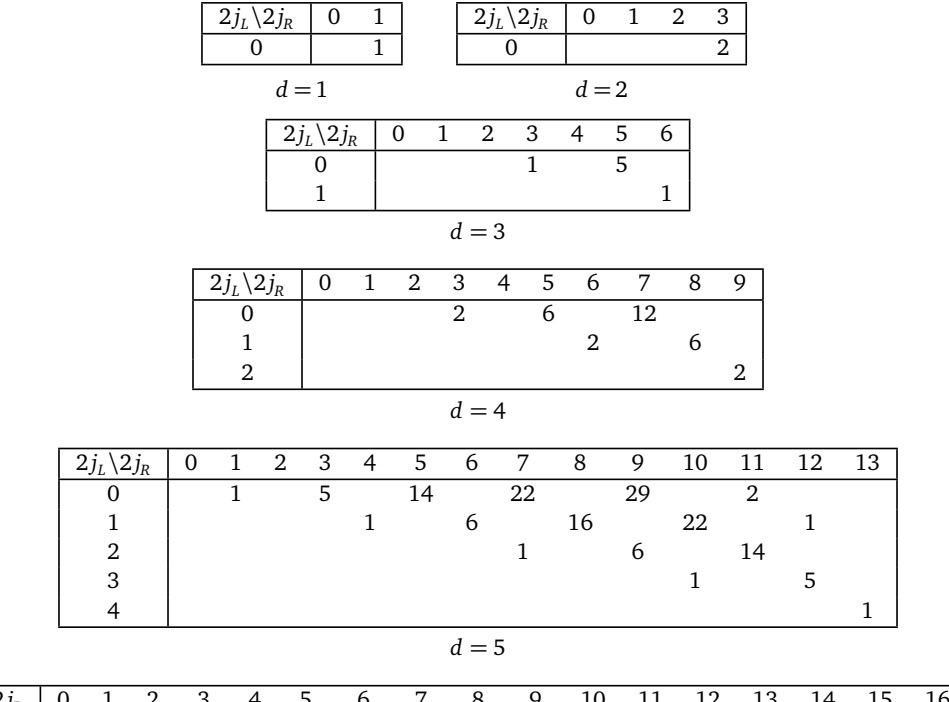

| $2j_L \backslash 2j_R$ | 0 | 1 |
| --- | --- | --- |
| 0 | | 1 |

$d = 1$

| $2j_L \backslash 2j_R$ | 0 | 1 | 2 | 3 |
| --- | --- | --- | --- | --- |
| 0 | | | | 2 |

$d = 2$

| $2j_L \backslash 2j_R$ | 0 | 1 | 2 | 3 | 4 | 5 | 6 |
| --- | --- | --- | --- | --- | --- | --- | --- |
| 0 | | | | 1 | | 5 | |
| 1 | | | | | | | 1 |

$d = 3$

| $2j_L \backslash 2j_R$ | 0 | 1 | 2 | 3 | 4 | 5 | 6 | 7 | 8 | 9 |
| --- | --- | --- | --- | --- | --- | --- | --- | --- | --- | --- |
| 0 | | | | 2 | | 6 | | 12 | | |
| 1 | | | | | | | 2 | | 6 | |
| 2 | | | | | | | | | | 2 |

$d = 4$

| $2j_L \backslash 2j_R$ | 0 | 1 | 2 | 3 | 4 | 5 | 6 | 7 | 8 | 9 | 10 | 11 | 12 | 13 |
| --- | --- | --- | --- | --- | --- | --- | --- | --- | --- | --- | --- | --- | --- | --- |
| 0 | | 1 | | 5 | | 14 | | 22 | | 29 | | 2 | | |
| 1 | | | | | 1 | | 6 | | 16 | | 22 | | 1 | |
| 2 | | | | | | | | 1 | | 6 | | 14 | | |
| 3 | | | | | | | | | | | 1 | | 5 | |
| 4 | | | | | | | | | | | | | | 1 |

$d = 5$

| $2j_L \backslash 2j_R$ | 0 | 1 | 2 | 3 | 4 | 5 | 6 | 7 | 8 | 9 | 10 | 11 | 12 | 13 | 14 | 15 | 16 | 17 |
| --- | --- | --- | --- | --- | --- | --- | --- | --- | --- | --- | --- | --- | --- | --- | --- | --- | --- | --- |
| 0 | | 6 | | 18 | | 36 | | 60 | | 74 | | 78 | | 14 | | 2 | | |
| 1 | | | 2 | | 8 | | 24 | | 50 | | 74 | | 76 | | 14 | | | |
| 2 | | | | | | 2 | | 8 | | 26 | | 50 | | 60 | | 6 | | |
| 3 | | | | | | | | | 2 | | 8 | | 24 | | 36 | | 2 | |
| 4 | | | | | | | | | | | | 2 | | 8 | | 18 | | |
| 5 | | | | | | | | | | | | | | | 2 | | 6 | |
| 6 | | | | | | | | | | | | | | | | | | 2 |

$d = 6$

Table 14: BPS spectrum of the Wilson loop for local $\mathbb{P}^1 \times \mathbb{P}^1$ with $\mathsf{n} = 2$ and $d \leq 6$.

| $2j_L \backslash 2j_R$ | 0 |
|---|---|
| 0 | 1 |

$d = \frac{1}{2}$

| $2j_L \backslash 2j_R$ | 0 | 1 | 2 |
|---|---|---|---|
| 0 | | | 2 |

$d = \frac{3}{2}$

| $2j_L \backslash 2j_R$ | 0 | 1 | 2 | 3 | 4 | 5 |
|---|---|---|---|---|---|---|
| 0 | | | 1 | | 6 | |
| 1 | | | | | | 1 |

$d = \frac{5}{2}$

| $2j_L \backslash 2j_R$ | 0 | 1 | 2 | 3 | 4 | 5 | 6 | 7 | 8 |
|---|---|---|---|---|---|---|---|---|---|
| 0 | | | 2 | | 8 | | 18 | | |
| 1 | | | | | | 2 | | 8 | |
| 2 | | | | | | | | | 2 |

$d = \frac{7}{2}$

| $2j_L \backslash 2j_R$ | 0 | 1 | 2 | 3 | 4 | 5 | 6 | 7 | 8 | 9 | 10 | 11 | 12 |
|---|---|---|---|---|---|---|---|---|---|---|---|---|---|
| 0 | 1 | | 6 | | 20 | | 38 | | 51 | | 3 | | |
| 1 | | | | 1 | | 7 | | 23 | | 38 | | 1 | |
| 2 | | | | | | | 1 | | 7 | | 20 | | |
| 3 | | | | | | | | | | 1 | | 6 | |
| 4 | | | | | | | | | | | | | 1 |

$d = \frac{9}{2}$

| $2j_L \backslash 2j_R$ | 0 | 1 | 2 | 3 | 4 | 5 | 6 | 7 | 8 | 9 | 10 | 11 | 12 | 13 | 14 | 15 | 16 |
|---|---|---|---|---|---|---|---|---|---|---|---|---|---|---|---|---|---|
| 0 | 6 | | 26 | | 60 | | 110 | | 148 | | 154 | | 28 | | 2 | | |
| 1 | | 2 | | 10 | | 34 | | 82 | | 138 | | 150 | | 22 | | | |
| 2 | | | | | 2 | | 10 | | 36 | | 82 | | 110 | | 8 | | |
| 3 | | | | | | | | 2 | | 10 | | 34 | | 60 | | 2 | |
| 4 | | | | | | | | | | | 2 | | 10 | | 26 | | |
| 5 | | | | | | | | | | | | | | 2 | | 8 | |
| 6 | | | | | | | | | | | | | | | | | 2 |

$d = \frac{11}{2}$

Table 15: BPS spectrum of the Wilson loop for local $\mathbb{P}^1 \times \mathbb{P}^1$ with $\mathsf{n} = 3$ and $d \leq \frac{11}{2}$.

| $2j_L\backslash 2j_R$ | 0 | 1 |
|---|---|---|
| 0 |  | 2 |

$d = 1$

| $2j_L\backslash 2j_R$ | 0 | 1 | 2 | 3 | 4 |
|---|---|---|---|---|---|
| 0 |  | 1 |  | 7 |  |
| 1 |  |  |  |  | 1 |

$d = 2$

| $2j_L\backslash 2j_R$ | 0 | 1 | 2 | 3 | 4 | 5 | 6 | 7 |
|---|---|---|---|---|---|---|---|---|
| 0 |  | 2 |  | 10 |  | 26 |  |  |
| 1 |  |  |  |  | 2 |  | 10 |  |
| 2 |  |  |  |  |  |  |  | 2 |

$d = 3$

| $2j_L\backslash 2j_R$ | 0 | 1 | 2 | 3 | 4 | 5 | 6 | 7 | 8 | 9 | 10 | 11 |
|---|---|---|---|---|---|---|---|---|---|---|---|---|
| 0 |  | 7 |  | 27 |  | 61 |  | 89 |  | 4 |  |  |
| 1 |  |  | 1 |  | 8 |  | 31 |  | 61 |  | 1 |  |
| 2 |  |  |  |  |  | 1 |  | 8 |  | 27 |  |  |
| 3 |  |  |  |  |  |  |  |  | 1 |  | 7 |  |
| 4 |  |  |  |  |  |  |  |  |  |  |  | 1 |

$d = 4$

| $2j_L\backslash 2j_R$ | 0 | 1 | 2 | 3 | 4 | 5 | 6 | 7 | 8 | 9 | 10 | 11 | 12 | 13 | 14 | 15 |
|---|---|---|---|---|---|---|---|---|---|---|---|---|---|---|---|---|
| 0 |  | 34 |  | 94 |  | 192 |  | 286 |  | 304 |  | 50 |  | 2 |  |  |
| 1 | 2 |  | 12 |  | 46 |  | 126 |  | 242 |  | 288 |  | 32 |  |  |  |
| 2 |  |  |  | 2 |  | 12 |  | 48 |  | 126 |  | 192 |  | 10 |  |  |
| 3 |  |  |  |  |  |  | 2 |  | 12 |  | 46 |  | 94 |  | 2 |  |
| 4 |  |  |  |  |  |  |  |  |  | 2 |  | 12 |  | 36 |  |  |
| 5 |  |  |  |  |  |  |  |  |  |  |  |  | 2 |  | 10 |  |
| 6 |  |  |  |  |  |  |  |  |  |  |  |  |  |  |  | 2 |

$d = 5$

Table 16: BPS spectrum of the Wilson loop for local $\mathbb{P}^1 \times \mathbb{P}^1$ with $\mathsf{n} = 4$ and $d \leq 5$.

$d = \frac{1}{2}$

| $2j_L\backslash 2j_R$ | 0 |
|---|---|
| 0 | 2 |

$d = \frac{3}{2}$

| $2j_L\backslash 2j_R$ | 0 | 1 | 2 | 3 |
|---|---|---|---|---|
| 0 | 1 |  | 8 |  |
| 1 |  |  |  | 1 |

$d = \frac{5}{2}$

| $2j_L\backslash 2j_R$ | 0 | 1 | 2 | 3 | 4 | 5 | 6 |
|---|---|---|---|---|---|---|---|
| 0 | 2 |  | 12 |  | 36 |  |  |
| 1 |  |  |  | 2 |  | 12 |  |
| 2 |  |  |  |  |  |  | 2 |

$d = \frac{7}{2}$

| $2j_L\backslash 2j_R$ | 0 | 1 | 2 | 3 | 4 | 5 | 6 | 7 | 8 | 9 | 10 |
|---|---|---|---|---|---|---|---|---|---|---|---|
| 0 | 7 |  | 35 |  | 92 |  | 150 |  | 5 |  |  |
| 1 |  | 1 |  | 9 |  | 40 |  | 92 |  | 1 |  |
| 2 |  |  |  |  | 1 |  | 9 |  | 35 |  |  |
| 3 |  |  |  |  |  |  |  | 1 |  | 8 |  |
| 4 |  |  |  |  |  |  |  |  |  |  | 1 |

$d = \frac{9}{2}$

| $2j_L\backslash 2j_R$ | 0 | 1 | 2 | 3 | 4 | 5 | 6 | 7 | 8 | 9 | 10 | 11 | 12 | 13 | 14 |
|---|---|---|---|---|---|---|---|---|---|---|---|---|---|---|---|
| 0 | 36 |  | 138 |  | 318 |  | 528 |  | 592 |  | 82 |  | 2 |  |  |
| 1 |  | 14 |  | 60 |  | 184 |  | 400 |  | 530 |  | 44 |  |  |  |
| 2 |  |  | 2 |  | 14 |  | 62 |  | 184 |  | 318 |  | 12 |  |  |
| 3 |  |  |  |  |  | 2 |  | 14 |  | 60 |  | 140 |  | 2 |  |
| 4 |  |  |  |  |  |  |  |  | 2 |  | 14 |  | 48 |  |  |
| 5 |  |  |  |  |  |  |  |  |  |  |  | 2 |  | 12 |  |
| 6 |  |  |  |  |  |  |  |  |  |  |  |  |  |  | 2 |

Table 17: BPS spectrum of the Wilson loop for local $\mathbb{P}^1 \times \mathbb{P}^1$ with $\mathsf{n} = 5$ and $d \leq \frac{9}{2}$.

$d = 1$

| $2j_L\backslash 2j_R$ | 0 | 1 | 2 |
|---|---|---|---|
| 0 |  | 9 |  |
| 1 |  |  | 1 |

$d = 2$

| $2j_L\backslash 2j_R$ | 0 | 1 | 2 | 3 | 4 | 5 |
|---|---|---|---|---|---|---|
| 0 |  | 14 |  | 48 |  |  |
| 1 |  |  | 2 |  | 14 |  |
| 2 |  |  |  |  |  | 2 |

$d = 3$

| $2j_L\backslash 2j_R$ | 0 | 1 | 2 | 3 | 4 | 5 | 6 | 7 | 8 | 9 |
|---|---|---|---|---|---|---|---|---|---|---|
| 0 |  | 43 |  | 132 |  | 242 |  | 6 |  |  |
| 1 | 1 |  | 10 |  | 50 |  | 132 |  | 1 |  |
| 2 |  |  |  | 1 |  | 10 |  | 44 |  |  |
| 3 |  |  |  |  |  |  | 1 |  | 9 |  |
| 4 |  |  |  |  |  |  |  |  |  | 1 |

$d = 4$

| $2j_L\backslash 2j_R$ | 0 | 1 | 2 | 3 | 4 | 5 | 6 | 7 | 8 | 9 | 10 | 11 | 12 | 13 |
|---|---|---|---|---|---|---|---|---|---|---|---|---|---|---|
| 0 |  | 186 |  | 500 |  | 928 |  | 1122 |  | 126 |  | 2 |  |  |
| 1 | 14 |  | 76 |  | 258 |  | 628 |  | 930 |  | 58 |  |  |  |
| 2 |  |  |  | 2 |  | 16 |  | 78 |  | 258 |  | 502 |  | 14 |
| 3 |  |  |  |  | 2 |  | 16 |  | 76 |  | 200 |  | 2 |  |
| 4 |  |  |  |  |  |  |  | 2 |  | 16 |  | 62 |  |  |
| 5 |  |  |  |  |  |  |  |  |  |  | 2 |  | 14 |  |
| 6 |  |  |  |  |  |  |  |  |  |  |  |  |  | 2 |

Table 18: BPS spectrum of the Wilson loop for local $\mathbb{P}^1 \times \mathbb{P}^1$ with $\mathsf{n} = 6$ and $d \leq 4$.

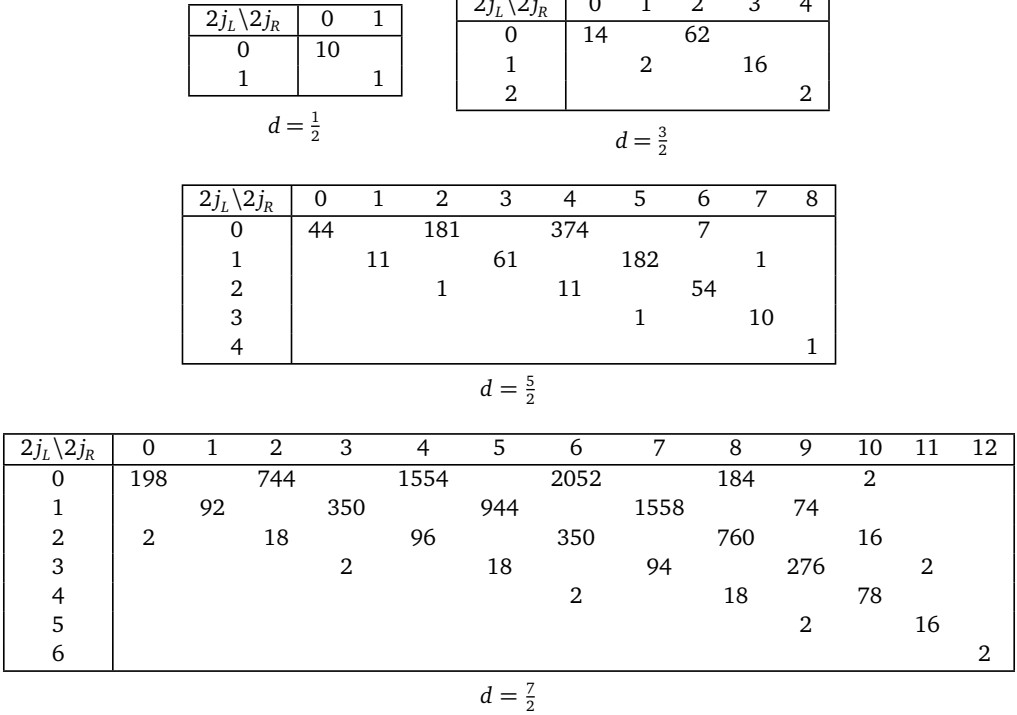

**$d = \tfrac{1}{2}$**

| $2j_L\backslash 2j_R$ | 0 | 1 |
|---|---|---|
| 0 | 10 | |
| 1 | | 1 |

**$d = \tfrac{3}{2}$**

| $2j_L\backslash 2j_R$ | 0 | 1 | 2 | 3 | 4 |
|---|---|---|---|---|---|
| 0 | 14 | | 62 | | |
| 1 | | 2 | | 16 | |
| 2 | | | | | 2 |

**$d = \tfrac{5}{2}$**

| $2j_L\backslash 2j_R$ | 0 | 1 | 2 | 3 | 4 | 5 | 6 | 7 | 8 |
|---|---|---|---|---|---|---|---|---|---|
| 0 | 44 | | 181 | | 374 | | 7 | | |
| 1 | | 11 | | 61 | | 182 | | 1 | |
| 2 | | | 1 | | 11 | | 54 | | |
| 3 | | | | | | 1 | | 10 | |
| 4 | | | | | | | | | 1 |

**$d = \tfrac{7}{2}$**

| $2j_L\backslash 2j_R$ | 0 | 1 | 2 | 3 | 4 | 5 | 6 | 7 | 8 | 9 | 10 | 11 | 12 |
|---|---|---|---|---|---|---|---|---|---|---|---|---|---|
| 0 | 198 | | 744 | | 1554 | | 2052 | | 184 | | 2 | | |
| 1 | | 92 | | 350 | | 944 | | 1558 | | 74 | | | |
| 2 | 2 | | 18 | | 96 | | 350 | | 760 | | 16 | | |
| 3 | | | | 2 | | 18 | | 94 | | 276 | | 2 | |
| 4 | | | | | | | 2 | | 18 | | 78 | | |
| 5 | | | | | | | | | | 2 | | 16 | |
| 6 | | | | | | | | | | | | | 2 |

Table 19: BPS spectrum of the Wilson loop for local $\mathbb{P}^1 \times \mathbb{P}^1$ with n = 7 and $d \leq \tfrac{7}{2}$.

**$d = 1$**

| $2j_L\backslash 2j_R$ | 0 | 1 | 2 | 3 |
|---|---|---|---|---|
| 0 | | 78 | | |
| 1 | 2 | | 18 | |
| 2 | | | | 2 |

**$d = 2$**

| $2j_L\backslash 2j_R$ | 0 | 1 | 2 | 3 | 4 | 5 | 6 | 7 |
|---|---|---|---|---|---|---|---|---|
| 0 | | 232 | | 556 | | 8 | | |
| 1 | 11 | | 73 | | 243 | | 1 | |
| 2 | | 1 | | 12 | | 65 | | |
| 3 | | | | | 1 | | 11 | |
| 4 | | | | | | | | 1 |

**$d = 3$**

| $2j_L\backslash 2j_R$ | 0 | 1 | 2 | 3 | 4 | 5 | 6 | 7 | 8 | 9 | 10 | 11 |
|---|---|---|---|---|---|---|---|---|---|---|---|---|
| 0 | | 1014 | | 2482 | | 3610 | | 258 | | 2 | | |
| 1 | 94 | | 460 | | 1368 | | 2502 | | 92 | | | |
| 2 | | 20 | | 116 | | 462 | | 1110 | | 18 | | |
| 3 | | | 2 | | 20 | | 114 | | 370 | | 2 | |
| 4 | | | | | | 2 | | 20 | | 96 | | |
| 5 | | | | | | | | | 2 | | 18 | |
| 6 | | | | | | | | | | | | 2 |

Table 20: BPS spectrum of the Wilson loop for local $\mathbb{P}^1 \times \mathbb{P}^1$ with n = 8 and $d \leq 3$.

| $2j_L\backslash 2j_R$ | 0 | 1 | 2 |
|---|---|---|---|
| 0 | 96 | | |
| 1 | | 20 | |
| 2 | | | 2 |

$$d=\tfrac{1}{2}$$

| $2j_L\backslash 2j_R$ | 0 | 1 | 2 | 3 | 4 | 5 | 6 |
|---|---|---|---|---|---|---|---|
| 0 | 240 | | 799 | | 9 | | |
| 1 | | 85 | | 316 | | 1 | |
| 2 | 1 | | 13 | | 77 | | |
| 3 | | | | 1 | | 12 | |
| 4 | | | | | | | 1 |

$$d=\tfrac{3}{2}$$

| $2j_L\backslash 2j_R$ | 0 | 1 | 2 | 3 | 4 | 5 | 6 | 7 | 8 | 9 | 10 |
|---|---|---|---|---|---|---|---|---|---|---|---|
| 0 | 1088 | | 3754 | | 6112 | | 350 | | 2 | | |
| 1 | | 574 | | 1920 | | 3870 | | 112 | | | |
| 2 | 20 | | 138 | | 596 | | 1572 | | 20 | | |
| 3 | | 2 | | 22 | | 136 | | 484 | | 2 | |
| 4 | | | | | 2 | | 22 | | 116 | | |
| 5 | | | | | | | | 2 | | 20 | |
| 6 | | | | | | | | | | | 2 |

$$d=\tfrac{5}{2}$$

Table 21: BPS spectrum of the Wilson loop for local $\mathbb{P}^1\times\mathbb{P}^1$ with $n=9$ and $d\leq\tfrac{5}{2}$.

## C.3 Refined BPS invariants for $E_n$ del Pezzos

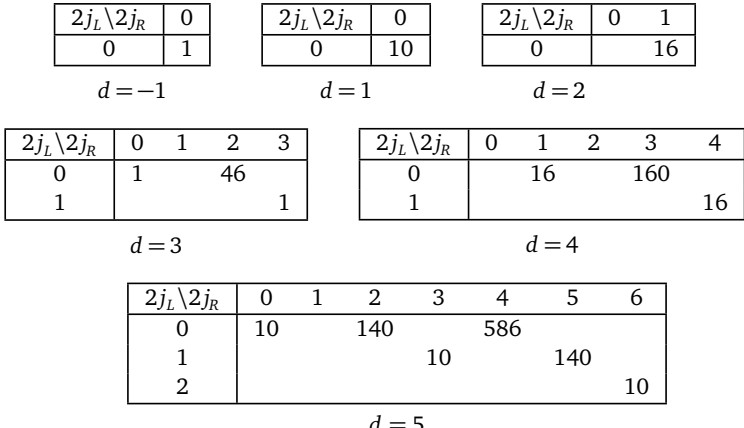

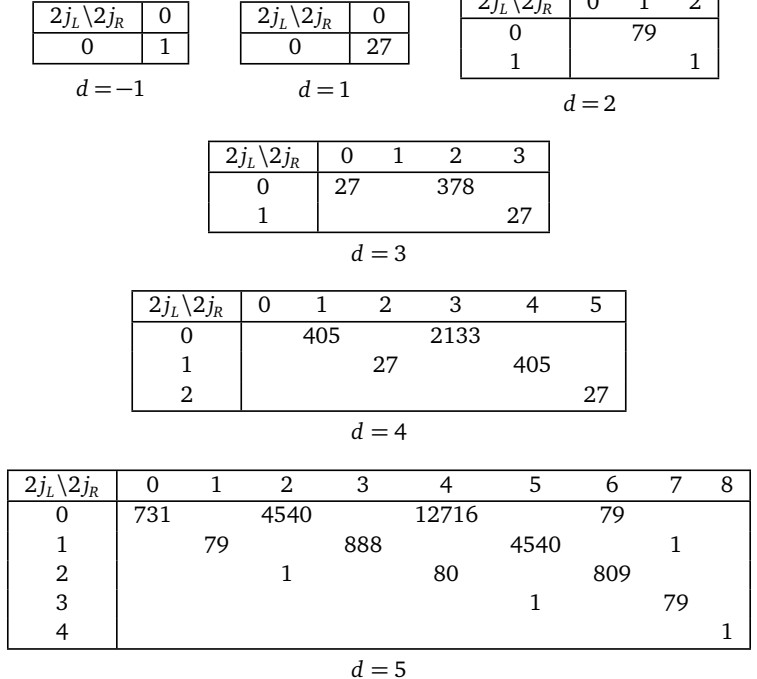

Table 22: BPS spectrum of the Wilson loop in the fundamental representation for $D_5$ del Pezzo.

Table 23: BPS spectrum of the Wilson loop in the fundamental representation for $E_6$ del Pezzo.

**d = −1**

| 2j_L\2j_R | 0 |
|---|---|
| 0 | 1 |

**d = 1**

| 2j_L\2j_R | 0 | 1 |
|---|---|---|
| 0 | 134 | |
| 1 | | 1 |

**d = 2**

| 2j_L\2j_R | 0 | 1 | 2 |
|---|---|---|---|
| 0 | | 968 | |
| 1 | | | 56 |

**d = 3**

| 2j_L\2j_R | 0 | 1 | 2 | 3 | 4 | 5 |
|---|---|---|---|---|---|---|
| 0 | 1674 | 10451 | 1807 | | | |
| 1 | | 134 | 1 | | | |
| 2 | | | 134 | 1 | | |
| 3 | | | | | | 1 |

**d = 4**

| 2j_L\2j_R | 0 | 1 | 2 | 3 | 4 | 5 | 6 | 7 |
|---|---|---|---|---|---|---|---|---|
| 0 | | 43640 | 129768 | 44552 | 1024 | | | |
| 1 | 968 | | 9496 | 1080 | | 56 | | |
| 2 | | | 56 | | 8528 | | | |
| 3 | | | | | | | 1024 | |
| 4 | | | | | | | | 56 |

**d = 5**

| 2j_L\2j_R | 0 | 1 | 2 | 3 | 4 | 5 | 6 | 7 | 8 | 9 | 10 | 11 |
|---|---|---|---|---|---|---|---|---|---|---|---|---|
| 0 | 237827 | 930381 | 1712249 | 939294 | 310203 | 68140 | 15529 | 1942 | 134 | 1 | | |
| 1 | 1808 | 86097 | 376669 | 91175 | 15797 | 1943 | 135 | 1 | | | | |
| 2 | | 15796 | | 75779 | | 310203 | | 13856 | | 1808 | | 134 |
| 3 | | | 1943 | | 1943 | | 1943 | | 135 | | 134 | |
| 4 | | | 135 | | 135 | | 135 | | 1 | | | |
| 5 | | | 1 | | 1 | | 1 | | | | | 1 |

Table 24: BPS spectrum of the Wilson loop in the fundamental representation for $E_7$ del Pezzo.

**$d = -1$**

| $2j_L \backslash 2j_R$ | 0 |
|---|---|
| 0 | 1 |

**$d = 1$**

| $2j_L \backslash 2j_R$ | 0 | 1 | 2 |
|---|---|---|---|
| 0 | 4125 | | |
| 1 | | 249 | |
| 2 | | | 1 |

**$d = 2$**

| $2j_L \backslash 2j_R$ | 0 | 1 | 2 | 3 | 4 | 5 |
|---|---|---|---|---|---|---|
| 0 | | 186126 | | 249 | | |
| 1 | 4124 | | 38877 | | 1 | |
| 2 | | 249 | | 4373 | | |
| 3 | | | 1 | | 249 | |
| 4 | | | | | | 1 |

**$d = 3$**

| $2j_L \backslash 2j_R$ | 0 | 1 | 2 | 3 | 4 | 5 | 6 | 7 | 8 | 9 |
|---|---|---|---|---|---|---|---|---|---|---|
| 0 | 3694119 | | 11393622 | | 4880618 | | 252004 | | 43498 | |
| 1 | | 1434130 | | 1286881 | | 295005 | | 43747 | | 249 |
| 2 | | | 39125 | | 256377 | | 4623 | | 250 | |
| 3 | | | | 4622 | | 39374 | | 250 | | 1 |
| 4 | | | | | 1 | | 4374 | | 1 | |
| 5 | | | | | | | | 250 | | |
| 6 | | | | | | | | | 250 | |
| 7 | | | | | | | | | | 1 |

**$d = 4$**

| $2j_L \backslash 2j_R$ | 0 | 1 | 2 | 3 | 4 | 5 | 6 | 7 | 8 | 9 | 10 | 11 | 12 | 13 | 14 |
|---|---|---|---|---|---|---|---|---|---|---|---|---|---|---|---|
| 0 | | 485875765 | | 799689237 | | 285477872 | | 72308485 | | | | | | | |
| 1 | 72035619 | | 292038369 | | 545982777 | | 7613632 | | 1649255 | | | | | | |
| 2 | | 29699120 | | 106691237 | | 234719994 | | 1357379 | | 300623 | | | | | |
| 3 | 1387759 | | 7873883 | | 30967129 | | 80571370 | | 300373 | | 250 | | | | |
| 4 | | 300621 | | 1692754 | | 7688504 | | 23680624 | | 44247 | | 1 | | | |
| 5 | | | 4375 | | 305246 | | 1649753 | | 6079371 | | 4625 | | | | |
| 6 | | | | 250 | | 44248 | | 300624 | | 1353506 | | 250 | | | |
| 7 | | | | | | | 4625 | | 43998 | | 256875 | | | | |
| 8 | | | | | | | | 250 | | 4624 | | 39375 | | | |
| 9 | | | | | | | | | 1 | | 250 | | 4374 | | |
| 10 | | | | | | | | | | | | 1 | | 250 | |
| 11 | | | | | | | | | | | | | | | 1 |

Table 25: BPS spectrum of the Wilson loop in the fundamental representation for $E_8$ del Pezzo.

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
