# Peer review of "Wilson loops, holomorphic anomaly equations and blowup equations"

_SciPost Physics_

## Round 1 · Referee Report · Anonymous (Referee 1) · 2023-6-17

Strengths

  1. Develops efficient computational techniques
  2. Proposes an extension of the blow-up equations

Weaknesses

  1. Lack of clarity at certain passages

Report

The paper extends the study on Wilson loops in five-dimensional supersymmetric field theories and their relations with topological strings. An interesting family of refined BPS invariants, which are associated to non-compact Calabi-Yau geometries, are introduced and computed. The main result of the paper is the powerful computational technique developed in Section 3. The work also introduces a nice proposal in Section 5 to extend the blow-up equations of refs. [38-40] beyond their original regime of validity. The results of the paper are of interest for the communities working of various related areas.

However, there are two main issues that must be addressed, namely Equation (43) and Section 4. - Equation (43) is the core of the paper but it has some unexplained parameter $\mathsf{n}^{\prime}$ on its right-hand side, that is not expected to be there, it is not defined, and does not appear anywhere else. - While the computations in Section 4 seem to be correct, the entire setup is unclear and should be explained. These and additional corrections to be made are listed in the "requested changes" section.

If the author satisfactorily addresses the main issues and the imprecisions listed below, I consider that the manuscript satisfies the acceptance criteria of SciPost Physics. If the authors corrects the major issues numbered (1), (3) and (5), but not (2) and (4), the paper would still have valuable content, and in that case I would recommend to be published in SciPost Core.

Requested changes

Major issues. 1) Equation (43) on page 12 is central for the result of the paper. However, the left-hand side only depends on the input $\mathsf{n}$ and on $(n,g)$, whereas the right-hand side depends on these three things, plus there is a sum over $(n^{\prime},g^{\prime})$. However,the summand on the right-hand side, also depends on a mysterious $\mathsf{n}^{\prime}$, which should not be there. How does it come about? It does not appear anywhere else. Equation (43) must be corrected and the results tabulated in Appendix C should be checked accordingly.

2) The discussion around Eq. (28) on page 9 is confusing. The Wilson loops arise from non-compact curves, while the Coulomb branch parameters of the 5d theory are associated to compact curves. As it is phrased now, the discussion in the lines just above (28) cannot be correct and should be improved.

3) Subsection 3.3, especially on the bottom of page 15: The author should clarify the notation conifold/orbifold/large volume point from the 5d gauge theory language. Is the confiold point the origin of the Coulomb branch? How is the orbifold point defined? What does "large volume" refer to, if $X$ is already non-compact? If "large volume" refers to the compact curve of which $t$ is the Kähler parameter, then it is the point at $\infty$ in the Coulomb branch.

4) Related to the previous point, the discussion at the beginning of Section 4 is phrased suitably for a 4D theory, but not for a 5D theory. If the conifold point is indeed the origin of the Coulomb branch, then the theory is strongly coupled there. It is not obvious to me why writing it as a theory of a 2-form gauge field would make it simpler. Is the ``dual Wilson loop'' mentioned below (87) a magnetic surface? If so, how is it placed in the 5D spacetime compatibly with the Omega background? If not, how can the magnetic dual of a Wilson loop in 5D be a loop?

5) Page 24 equation (120): How are $\mathbf{r}_a, \mathbf{r}_b$ on the right-hand side defined, exactly?

Imprecisions to be rectified/clarified. - Page 2: $SO(4)$ and $SU(2)\times SU(2)$ are not isomorphic as Lie groups. Since the author only uses $SU(2)_L \times SU(2)_R$ representations, I suggest any mentions to an "equality" with $SO(4)$ is removed. - Page 4 line below (6): the author writes that $Z_{\rm BPS}$ is defined in (2), but (2) is not really a definition, rather a non-trivial expansion. The definition as a Witten index is written in words above (2). - Page 5 two lines below (8): inserting the background non-compact curves --> inserting background M2 branes wrapping the non-compact curves. The curves are already in $X$. - Page 5 paragraph "The Wilson loops": By the relation $W_{\mathbf{r}1 \otimes \mathbf{r}_2}= W1}W$}_2 I understand that the setup includes an M2 brane wrapping each non-compact curve, while a single M2 wrapping all the curves would give $\oplus_i W_{\mathbf{r}_i}$. Whether this is the setup the author considers or not, it should be clearly stated. - Page 6 line 4: it is not obvious by the definition that $\widetilde{N}_{j_L,j_R}^{\beta}$ will be integers if $\mathsf{n}>0$. It should be discussed why this is the case. The author mentions this fact in the examples in Section 3.3, but it would be important to make things clear when the quantities are introduced. - Page 10 lines 6 and 7: $D_5$ and $E_6$ are not the same. - Page 10 line 10: the low energy theory is still a 5D theory, but it is not UV complete in 5D. Embedding in the E-string theory provides the UV completion. - Page 10 line 13: is $h$ associated to the generator of $H_2 (\mathbb{P}^2)$ or to the Poincare dual of $\mathbb{P}^2$ in $X$? It seems to be the latter, but it is unclear as it is written now.

Misprints. - Abstract, last line: satisfying --> satisfy - page 2 before (1): inserting --> insertion of - page 2 before (2): topological invariant which counts --> topological invariants which count - page 4 line 8: generalization --> generalizations - page 4 two lines before (7): Why is $\mathcal{E}$ called singular part? - page 6 line before (14): the acronym BCOV should be defined. - page 8 line 1: right-hand side sum --> right-hand side sums. - page 8 between (24) and (25): notation --> definition. - page 8 line 14: low --> lower - page 9 the sentence: Lastly, we emphasize that the initial conditions for the BPS sectors are completely the choice by hand --> rephrase/clarify. - page 11: the $(n + 1)$-point blow up of $\mathbb{P}^2$ is isomorphic to the $n$-point blow up of $\mathbb{P}^1 \times \mathbb{P}^1$ --> the $n$-point blow up of $\mathbb{P}^2$ is isomorphic to the $(n-1)$-point blow up of $\mathbb{P}^1 \times \mathbb{P}^1$. - page 11 sentece after the previous, $N_f=n$ --> $N_f=n-1$. - page 12 below (44): Christoffel symbol --> Christoffel symbol on the complex structure moduli space of $X$. - titles of subsections 3.3.1 and 3.3.2: local --> Local - page 22 paragraph "Example": whose geometry is corresponding --> whose geometry corresponds - page 23 the sentence: among all the physical observables, the Wilson loop partition functions satisfy all the properties --> rephrase/clarify. page 23 line above (108): negative $Q$ term --> negative powers of $Q$

---

## Round 1 · Referee Report · Anonymous (Referee 2) · 2023-7-7

Strengths

1- The author writes down the holomorphic anomaly equations for the BPS sectors of Wilson loop expectation values for refined topological string on non-compact Calabi-Yau threefolds, which is technically superior than the holomorphic anomaly equations for the Wilson loop amplitudes. 2- The author performs detailed calculations of the BPS sectors of Wilson loop expectation values of the local $\mathbb{P}^2$ model and the local $\mathbb{P}^1\times \mathbb{P}^1$ model, and proceeded to extract the related BPS invariants, generating a lot of data, which beg for geometric interpretation. 3- The author proposes a much generalised version of the blowup equations for the refined partition function of topological string, with or without Wilson loop insertion, in principle for any B-field that satisfies the flux quantisation condition, greatly enlarging the catalogue of blowup equations.

Weaknesses

1- One of the major results of the paper, the holomorphic anomaly equations for the BPS sectors of Wilson loop expectation values, is more-or-less a rewriting of the holomorphic anomaly equations for the Wilson loop amplitudes in the author's previous paper arXiv:2205.02366. The author claims the former is technically more powerful than the latter, but the reason of so is unclear. 2- The solution of perturbative energy spectra of quantum mirror curves using Wilson loops, proposed in section four, as the author also admits, is in principle no different from the author's previous paper arXiv:1406.6178. Arguably the method here is even more complicated than those in arXiv:1406.6178, involving the heavy machinery of holomorphic anomaly equations. 3- Section five on generalised blowup equations is not fully developed. In particular, the most generic form of the blowup equations is proposed in section 5.3 without any tests or checks. 4- The author could use much improvement in the English writing.

Report

In this paper, the author investigates the computation of the expectation values of half-BPS Wilson loops in five dimensional $N=1$ supersymmetric field theories with Omega background, which are geometrically engineered by refined topological string theory whose target spaces are non-compact Calabi-Yau threefolds.

Instead of working on the Wilson loops themselves, the author proposes to focus on the BPS sectors, which are more-or-less indecomposable components of the Wilson loop expectation values. The author derives the holomorphic anomaly equations for the BPS sectors, and solves them extensively using the direct integration methods for two exemplary models, the local $\mathbb{P}^2$ and the local $\mathbb{P}^1\times \mathbb{P}^1$. The author proceeds to extract a great many associated BPS invariants from the BPS sectors. These are very interesting geometric invariants, which are begging for precise mathematical interpretations.

On the other hand, it seems that the holomorphic anomaly equations for the BPS sectors are a simple re-writing of the holomorphic anomaly equations for the Wilson loop amplitudes, which are logarithms of Wilson loop expectation values, in the author's previous paper arXiv:2205.02366, where many examples of Wilson loop amplitudes as well as Wilson loop BPS invariants have already been computed. The author claims the current method is technically more powerful than those in arXiv:2205.02366. If so, the reason is not quite clear.

Next, the author proposes to solve the perturbative energy spectra of quantum mirror curves associated to non-compact Calabi-Yau threefolds using the Wilson loop expectation values.

However, as the author also admits, this method is in principle no different than those in the author's previous paper arXiv:1406.6178. Arguably, the method in this paper is technically even more complicated than arXiv:1406.6178, as it involves the heavy machinery of holomorphic anomaly equations, and it does not seem necessary for the relatively simple problem of solving the perturbative spectra of quantum mirror curves. The idea of computing energy spectra of quantum mirror curves using Wilson loop expectation values has in fact already appeared in the literature, for instance, in arXiv:1706.05142 and arXiv:1806.01407, which tackle with the much more difficult problems of computing the nonperturbative and exact energy spectra, and which the current paper also fails to cite.

Finally, using the data of Wilson loop expectation values, the author checks a very interesting generalisation of the blowup equations for refined topological string. Instead of a previously known bounded set of B-fields, the author proposes that one can always write down a blowup equation for any B-field that satisfies the flux quantisation condition, except that when B-field falls outside the bounded set, the Lambda factor is no longer a simple function of the mass parameters, but in principle can be a linear combination of Wilson loop expectation values. The author gives convincing arguments for this conjecture, and checks this conjecture with the example of local $\mathbb{P}^1\times\mathbb{P}^1$. This conjecture greatly enlarges the catalogue of blowup equations. And the author proceeds to propose a further generalisation of blowup equations for topological string partition functions with Wilson loop insertions. Unfortunately, this section does not look fully developed. In particular, the most generalised form of blowup equations is simply stated in section 5.3 without any tests or checks.

In terms of writing, the paper could use much improvement. There are some statements which are given without justification. For instance, the statement below (28) on page 9 that the higher genus parts of BPS sectors are ``hope''ed to be regular in the large volume limit. Are there any physical arguments for this statement or is it just the author's hope? There are also a lot of misprints, both in formulas and in words, as well as grammatical errors, which impair the readability of the paper. For instance, in the last sentence of the abstract, it should be "that satisfy" instead of "that satisfying". The author is advised to do a thorough proofreading to improve the manuscript.

In summary, the first half of the paper is a bit lacking in novelty as it is an improvement of the author's previous paper, and therefore is not a groundbreaking or pioneering work. The second half (section five) is genuinely novel, but it does not seem fully developed. The referee therefore recommends a major revision of the manuscript if it is to be accepted by this journal.

Requested changes

Major requests of changes 1- Maybe it is clear to the author, but could the author please explain why the boundary conditions for the BPS sectors are improved compared to the Wilson loop amplitudes, as stated for instance on page 7, which, as the author claims, makes the holomorphic anomaly equations for BPS sectors technically more powerful than those for Wilson loop amplitudes? In addition, this statement is checked for the local $\mathbb{P}^2$ model and the local $\mathbb{P}^1\times \mathbb{P}^1$ model. But is it universal for any non-compact Calabi-Yau threefold? 2- Please provide at least one simple example for the most generalised form of blowup equations in section 5.3. 3- Please do a thorough proofreading to reduce the amount of misprints and grammatical errors.

Minor requests of changes 1- On page 3 in (2), powers of $q_1, q_2$ should include the index $k$, and in addition, the index $k$ should be summed over. 2- On page 5 in (11), shouldn't the $M$s be tilded? 3- On page 8, the definition of the genus expansion $\mathcal{F}^{(n,g)}{\mathcal{S}$}} for the BPS sectors should be given before (23). This genus expansion is non-trivial and is different from the usual one as it seems to involve the number of insertions, as indicated by (40). 4- By comparing (23) on page 8 and (43) on page 13, it seems the subscript $\ mathsf{n}$ in the second F on the right hand side of (43) should be $\ mathsf{n}'$. There also misses a summation over $\mathsf{n'}$ in (43). In addition, if I use (40), I have difficulty in getting either (23) or (43). It seems in the second term on the right hand side, the total sum of genera and numbers of insertions should be the same as the left hand side, which is not met in either (23) or (43). Maybe the author could confirm or disprove. 5- On page 9 in the second paragraph, I think the roles of $b_4$ and $b_2$ should be exchanged. The same happens on page 22 in the first line. 6- On page 9 below (28), the author writes "we hope that the higher genus parts are regular under ..." This is not very scientific. As this condition is crucial for actually solving the BPS sectors, the author needs to clarify whether there is any justification for this requirement. 7- On page 10 in the first paragraph, it is not clear why the initial conditions can be chosen at will without affecting the final result. Maybe the author could clarify further. 8- On page 15 in (58), there should be $\partial_z$ in front of $C_{zzz}$. 9- On page 21 in (92), what are the values of $n,g$? Actually shouldn't the energy be identified with the NS Wilson loop instead of its components? 10- On page 22 above (99), I think $\alpha_i\rightarrow\infty$ should be $\alpha_i\rightarrow 0$, as indicated by (99). 11- On page 24 below (105), it is not clear why the author discusses the minimal value of $f_t$. Maybe the author could clarify. 12- On page 25 in (120), is there any relationship between the representations of Wilson loops on the right hand side and the possible representations of Wilson loops on the left hand side?

---

## Editorial Decision

awaiting_resubmission